# Federated Learning on Virtual Heterogeneous Data with Local-Global Dataset Distillation

**Chun-Yin Huang**                                    *chunyinh@ece.ubc.ca*
*University of British Columbia*
*Vector Institute*

**Ruinan Jin**                                    *ruinanjin@alumni.ubc.ca*
*University of British Columbia*
*Vector Institute*

**Can Zhao**                                    *canz@nvidia.com*
*NVIDIA*

**Daguang Xu**                                    *daguangx@nvidia.com*
*NVIDIA*

**Xiaoxiao Li**                                    *xiaoxiao.li@ece.ubc.ca*
*University of British Columbia*
*Vector Institute*

**Reviewed on OpenReview:** https://openreview.net/forum?id=QplBL2pV4Z

## Abstract

While Federated Learning (FL) is gaining popularity for training machine learning models in a decentralized fashion, numerous challenges persist, such as asynchronization, computational expenses, data heterogeneity, and gradient and membership privacy attacks. Lately, dataset distillation has emerged as a promising solution for addressing the aforementioned challenges by generating a compact synthetic dataset that preserves a model's training efficacy. *However, we discover that using distilled local datasets can amplify the heterogeneity issue in FL.* To address this, we propose **Fed**erated Learning on Virtual Heterogeneous Data with **L**ocal-**G**lobal **D**ataset **D**istillation (FEDLGD), where we seamlessly integrate dataset distillation algorithms into FL pipeline and train FL using a smaller synthetic dataset (referred as *virtual data*). Specifically, to harmonize the domain shifts, we propose iterative distribution matching to inpaint global information to *local virtual data* and use federated gradient matching to distill *global virtual data* that serve as anchor points to rectify heterogeneous local training, without compromising data privacy. We experiment on both benchmark and real-world datasets that contain heterogeneous data from different sources, and further scale up to an FL scenario that contains a large number of clients with heterogeneous and class-imbalanced data. Our method outperforms *state-of-the-art* heterogeneous FL algorithms under various settings. Our code is available at https://github.com/ubc-tea/FedLGD.

## 1 Introduction

Having a compatible training dataset is an essential *de facto* precondition in modern machine learning. However, in areas such as medical applications, collecting such a massive amount of data is not realistic since it may compromise privacy regulations such as GDPR (Voigt & Von dem Bussche, 2017). Thus, researchers seek to circumvent the privacy leakage by utilizing federated learning pipelines or training with synthetic data.

Federated learning (FL) (McMahan et al., 2017) has emerged as a pivotal paradigm for conducting machine learning on data from multiple sources in a distributed manner. Traditional FL involves a large number of clients collaborating to train a global model. By keeping data local and sharing only the local model updates, FL prevents the direct exposure of local datasets in collaborative training. However, despite these advantages, several research challenges remain in FL, including computational costs, asynchronization, data heterogeneity, and vulnerabilities to deep privacy attacks(Wen et al., 2023).

Another approach to GDPR compliance that has gained increased interest is using synthetic data in machine learning model training to supplement or replace real data when the latter is not suitable for direct use (Nikolenko, 2021). Among data synthesis methods, dataset distillation (Wang et al., 2018; Cazenavette et al., 2022; Zhao et al., 2021; Zhao & Bilen, 2021; 2023) has emerged as an ideal data synthesis strategy, as it is explored to enhance the efficiency and privacy of machine learning. Dataset distillation creates a compact synthetic dataset while retaining similar model performance of that trained on the original dataset, allowing efficiently training a machine learning model (Zhao et al., 2021; Zhao & Bilen, 2023). The distilled data usually remains low fidelity to the raw data but yet contains highly condensed essential information that makes the appearance of the distilled data dissimilar to the real data (Dong et al., 2022).

In this work, we introduce an effective training strategy that leverages both FL and virtual data generated via dataset distillation, referred as *federated virtual learning*, as the models are trained from virtual data (also referred as synthetic data) (Xiong et al., 2023; Goetz & Tewari, 2020; Hu et al., 2022). In particular, we aim to find the best way to incorporate dataset distillation into FL under ordinary FL pipeline, where the only change is replacing real data with virtual data for local training. A simple approach is to generate synthetic data first and then use it for FL training; however, this could lead to suboptimal performance in data heterogeneous settings. We observe increased divergence in loss curves in early FL rounds if we simply replace real data with distilled virtual data synthesized by Distribution Matching (DM) (Zhao & Bilen, 2023). Thus, We perform a simple experiment: Measure the distances between a set of digits datasets (please refer to `DIGITS` in Sec. 5.2 for details) before and after distillation with DM. Statistically, we find that the MMD scores increase after distillation, resulting in an averaged 37% increment. Empirically, we visualize the tSNE plots of two different datasets (USPS (Hull, 1994) and SynthDigits (Ganin & Lempitsky, 2015)) in Fig. 1, and distributions become diverse after distillation. This reveals that local virtual data from dataset distillation may worsen the data heterogeneity issue in FL. Note that the data heterogeneity referred here (also throughout the paper) is *domain shift*, which assumes variations in $P(X|y)$ across clients, where $X$ represents the input data and $y$ the corresponding labels. The concept differs from label shift ($P(y)$), which considers the heterogeneity on the labels.

To alleviate the problem of data heterogeneity, we enforce consistency in local embedded features using consensual anchors that capture global distribution. Existing works usually rely on the anchors yield from pre-generated noise (Tang et al., 2022) that cannot reflect training data distribution; or shared additional features from the client side (Zhou et al., 2023; Ye et al., 2023b), exposing more data leakage. To overcome the limitations, we propose an effective solution to address the heterogeneity issues using global virtual anchor for regularization, supported by our theoretical analysis. Without compromising privacy in implementation, our global anchors are distilled from **pre-existing** shared gradient information in FL to facilitate sharing global distribution information.

Apart from facilitating heterogeneous FL, such federated virtual learning further reduces computational cost and offers better empirical privacy protection. Specifically, we empirically demonstrate the

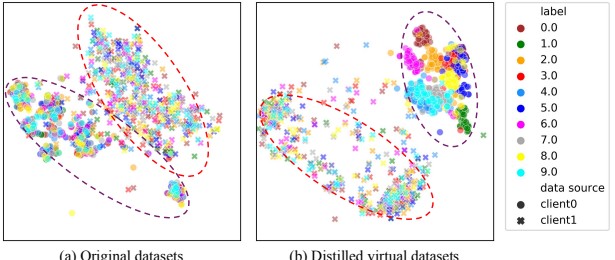

(a) Original datasets    (b) Distilled virtual datasets

Figure 1: Distilled local datasets using Distribution Matching (DM) (Zhao & Bilen, 2023) can worsen heterogeneity in FL. tSNE plots of (a) original datasets and (b) distilled virtual datasets of USPS (client 0) and SynthDigits (client 1). The two distributions are marked in the dashed curves. We observe fewer overlapped ∘ and × in (b) compared with (a), indicating higher heterogeneity between two clients after distillation. Statistically, we find that the Maximum Mean Discrepancy (MMD) (Gretton et al., 2012) scores increase after distillation, resulting in an averaged 37% increment.

reconstructed image from Gradient Inversion Attack (Geiping et al., 2020; Huang et al., 2021) trained on distilled data obtain much lower quality. We also present that our virtual training can defend better against Membership Inference Attacks (Shokri et al., 2017). Please refer to Sec. 5 for more details.

To this end, we propose FEDLGD, a federated virtual learning method with local and global distillation. Data distillation is gaining attention in centralized machine learning. Recognizing the need for efficiency in FL, we propose integrating two efficient dataset distillation methods into our FL pipeline. Specifically, we propose *iterative distribution matching* in local distillation by comparing the feature distribution of real and synthetic data using an evolving global feature extractor. The local distillation results in compact local virtual datasets with balanced class distributions, achieving efficiency and synchronization while avoiding class imbalance. In addition, unlike previously proposed federated virtual learning methods that rely solely on local distillation (Goetz & Tewari, 2020; Xiong et al., 2023; Hu et al., 2022), we also propose a novel and efficient method, *federated gradient matching*, that seamlessly integrate dataset distillation into FL pipeline to synthesize global virtual data as anchors on the server side using the uploaded averaged gradients. The global virtual data then serves as anchors to alleviate domain shifts among clients.

We conclude our contributions as follows:

- This paper focuses on an important but under-explored FL setting in which local models are trained on small virtual datasets, which we refer to as *federated virtual learning*, and we are *the first* to reveal that using distilled local virtual data using Distribution Matching (Zhao & Bilen, 2023) may *exacerbate* the heterogeneity problem in the federated virtual learning setting.

- We propose to address the heterogeneity problem by our novel distillation strategies, *iterative distribution matching* and *federated gradient matching*, that utilizes pre-existing shared information in FL, and theoretically show it can effectively lower the statistic margin.

- Through comprehensive experiments on benchmark and real-world datasets, we empirically show that FEDLGD outperforms existing state-of-the-art FL algorithms.

## 2 Related Work

### 2.1 Dataset Distillation

Data distillation (Wang et al., 2018) aims to improve data efficiency by distilling the most essential feature from a large-scale dataset (e.g., datasets comprising billions of data points) into a certain terse and high-fidelity dataset. For example, Gradient Matching (Zhao et al., 2021) is proposed to make the deep neural network produce similar gradients for both the terse synthetic images and the large-scale original dataset. Besides, (Cazenavette et al., 2022) proposes matching the model training trajectory between real and synthetic data to guide the update for distillation. Another popular way of conducting data distillation is through Distribution Matching (Zhao & Bilen, 2023). This strategy instead, attempts to match the distribution of the smaller synthetic dataset with the original large-scale dataset in latent representation space. It significantly improves the distillation efficiency. There are following works that further improves the utility of the distilled data (Li et al.; Zhang et al., 2024). Moreover, recent studies have justified that data distillation can defend against popular privacy attacks such as Gradient Inversion Attacks and Membership Inference Attacks (Dong et al., 2022; Carlini et al., 2022b), which is critical in federated learning. In practice, dataset distillation is used in healthcare for medical data sharing for privacy protection (Li et al., 2022). We refer the readers to (Sachdeva & McAuley, 2023) for further data distillation strategies.

### 2.2 Heterogeneous Federated Learning

FL performance downgrading on non-iid data is a critical challenge (Ye et al., 2023a). A variety of FL algorithms have been proposed ranging from global aggregation to local optimization to handle this heterogeneous issue, as echoed by (Sun et al., 2024), data heterogeneity plays a critical role for model generalization. *Global aggregation* improves the global model exchange process for better unitizing the updated client models to create a powerful server model. FedNova (Wang et al., 2020) notices an imbalance among different local

models caused by different levels of training stage (e.g., certain clients train more epochs than others) and tackles such imbalance by normalizing and scaling the local updates accordingly. FedRod (Chen & Chao, 2021) seeks to bridge personalized and generic FL by training separate global and local projection layers. Similarly, FedGELA Fan et al. (2024) also aims to bridge personalized and generic FL, employing simplex equiangular tight frame (ETF) to address class-imbalance data. Meanwhile, FedAvgM (Hsu et al., 2019) applies the momentum to server model aggregation to stabilize the optimization. Furthermore, there are strategies to refine the server model or client models using knowledge distillation such as FedDF (Lin et al., 2020), FedGen (Zhu et al., 2021), FedFTG (Zhang et al., 2022), FedICT (Wu et al., 2023), FedGKT (He et al., 2020), FedDKC (Wu et al., 2022), and FedHKD (Chen et al., 2023). However, we consider knowledge distillation and data distillation two orthogonal directions to solve data heterogeneity issues. *Local training optimization* aims to explore the local objective to tackle the non-iid issue in FL system. FedProx (Li et al., 2020) straightly adds $L_2$ norm to regularize the client model and previous server model. Scaffold (Karimireddy et al., 2020) adds the variance reduction term to mitigate the "clients-drift". MOON (Li et al., 2021) brings mode-level contrastive learning to maximize the similarity between model representations to stable the local training. There is also another line of works (Ye et al., 2023b; Tang et al., 2022) proposed to use a global *anchor* to regularize local training. Global anchor can be either a set of virtual global data or global virtual representations in feature space. However, in (Tang et al., 2022), the empirical global anchor selection may not be suitable for data from arbitrary distribution as they don't update the anchor according to the training datasets. More recently, (Chen et al., 2024) propose to utilize communication compression to facilitate heterogeneous FL training. Other methods, such as those rely on feature sharing from clients (Zhou et al., 2023; Ye et al., 2023b), are less practical, as they pose greater data privacy risks compared to classical FL settings.

### 2.3 Datasets Distillation for FL

Dataset distillation for FL is an emerging topic that has attracted attention due to its benefit for efficient FL systems. It trains model on distilled synthetic datasets, thus we refer it as federated virtual learning. It can help with FL synchronization and improve training efficiency by condensing every client's data into a small set. To the best of our knowledge, there are few published works on distillation in FL. Concurrently with our work, some studies (Goetz & Tewari, 2020; Xiong et al., 2023; Hu et al., 2022; Huang et al., 2024) distill datasets locally and share the virtual datasets with other clients/servers. Although privacy is protected against *currently* existing attack models, we consider directly sharing local virtual data not a reliable strategy. It is worth noting that some recent works propose to share locally generated surrogates, such as prototypes (Tan et al., 2022), performance-sensitive features (Yang et al., 2024), or logits (Huang et al., 2024) instead of the global model parameters. However, this work focuses on combining dataset distillation with pre-existing shared information in the classical FL setting to alleviate the data heterogeneity problem.

## 3 Method

### 3.1 Setup for Federated Virtual Learning

We start with describing the classical FL setting. Suppose there are $N$ parties (clients) who own local datasets $(D^{c_1}, \ldots, D^{c_N})$, and the goal of a classical FL system, such as FedAvg (McMahan et al., 2017), is to train a global model with parameters $\theta$ on the distributed datasets $(D \equiv \bigcup_{i \in [N]} D^{c_i})$. The objective function is written as: $\mathcal{L}(\theta) = \sum_{i=1}^{N} \frac{|D^{c_i}|}{|D|} \mathcal{L}_i(\theta)$, where $\mathcal{L}_i(\theta)$ is the empirical loss of client $i$. In practice, different clients in FL may have variant amounts of training samples, leading to asynchronized updates. In this work, we focus on a new type of FL training method – federated virtual learning, that trains on virtual datasets for efficiency and synchronization (discussed in Sec.2.3). Federated virtual learning synthesizes local virtual data $\tilde{D}^{c_i}$ for client $i$ for $i \in [N]$ and form $\tilde{D} \equiv \bigcup_{i \in [N]} \tilde{D}^{c_i}$. Typically, $|\tilde{D}^{c_i}| \ll |D^{c_i}|$ and $|\tilde{D}^{c_i}| = |\tilde{D}^{c_j}|$. A basic setup for federated virtual learning is to replace $D^{c_i}$ with $\tilde{D}^{c_i}$ to train FL model on the virtual datasets.

Figure 2: Overview of the proposed method FEDLGD. We split FL rounds into *selected* and *unselected* rounds. For the *selected* rounds, clients will refine the local virtual data and update local models, while the server uses aggregated gradients to update global virtual data and the global model. We term this procedure Local-Global Data Distillation. For the *unselected* rounds, we perform ordinary FL training with virtual data while adding regularization loss on local model updating. In the middle box, we also show the evolution of global and virtual data. Observe that although local virtual does not change visually, we found the local distillation steps are essential for improving model performance as shown in Fig. 3c and 3d.

## 3.2 Overall Pipeline

The overall pipeline of our proposed method contains three phases, including *1) initialization, 2) local-global distillation, and 3) local-global model update.* We depict the essential design of FEDLGD in Fig. 2. We begin with the initialization of the clients' local virtual data $\tilde{D}^c$ by performing distribution matching (DM) (Zhao & Bilen, 2023). Meanwhile, the server will randomly initialize global virtual data $\tilde{D}^g$ and network parameters $\theta_0^g$. Then, we refine our local and global virtual data using our proposed *local-global* distillation strategies. Among the selected iterations, we update $\theta$, $\tilde{D}^g$, and $\tilde{D}^c$ in early training epochs, where the server and clients can update their virtual data to match global information. For the unselected iterations, we train $\theta$ using with additional regularization loss which penalizes the shift between local and global virtual data. The full algorithm is shown in Algorithm 1.

## 3.3 FL with Local-Global Dataset Distillation

### 3.3.1 Local Data Distillation for Federated Virtual Learning

First of all, we hope to distill virtual data conditional on class labels to achieve class-balanced virtual datasets. Second, we hope the virtual local data is best suited for the classification task. Last but not least, the process should be efficient due to the limited computational resource locally. To this end, we design Iterative Distribution Matching to fulfill our purpose.

**Iterative distribution matching.** The objective for this part is to gradually improve local distillation quality during FL. Given efficiency is critical for an FL system, we propose to adapt one of the most efficient yet effective data distillation method that leverage distribution matching (DM) in the representation space, DM (Zhao & Bilen, 2023), in an iterative updating form to be integrated with FL. To this end, we split

---

**Algorithm 1** Federated Virtual Learning with Local-global Distillation

---

**Require:** $f^\theta$: Model, $\psi^\theta$: Feature extractor, $\theta$: Model parameters, $\tilde{D}$: Virtual data, $D$: Original data, $\mathcal{L}$: Losses, $G$: Gradients.

1:
2: **Distillation Functions:**
3: $\quad \tilde{D}^c \leftarrow \mathrm{DM}(D^c, f^\theta)$  $\quad\quad\quad\quad\quad\quad\quad\quad\quad\quad\quad\quad\quad\quad\quad\quad\quad$ ▷ Distribution Matching
4: $\quad \tilde{D}_t^c \leftarrow \mathrm{IterativeDM}(\tilde{D}_{t-1}^c, f_t^\theta)$  $\quad\quad\quad\quad\quad\quad\quad\quad\quad\quad\quad$ ▷ Iterative Distribution Matching
5: $\quad \tilde{D}_{t+1}^g \leftarrow \mathrm{FederatedGM}(\tilde{D}_t^g, G_t^g)$  $\quad\quad\quad\quad\quad\quad\quad\quad\quad$ ▷ Federated Gradient Matching
6:
7: **Initialization:**
8: $\quad \tilde{D}_0^c \leftarrow \mathrm{DM}(D_{\mathrm{rand}}^c, f_{\mathrm{rand}}^\theta)$  $\quad\quad\quad\quad\quad\quad\quad$ ▷ Distilled local data for virtual FL training
9:
10: **FedLGD Pipeline:**
11: **for** $t = 1, \ldots, T$ **do**
12: $\quad$ **Clients:**
13: $\quad$ **for** each selected Client **do**
14: $\quad\quad$ **if** $t \in \tau$ **then**  $\quad\quad\quad\quad\quad\quad\quad\quad\quad\quad\quad\quad$ ▷ Local-global distillation
15: $\quad\quad\quad \tilde{D}_t^c \leftarrow \mathrm{IterativeDM}(\tilde{D}_{t-1}^c, f_t^\theta)$
16: $\quad\quad\quad G_t^c \leftarrow \nabla_\theta \mathcal{L}_{\mathrm{CE}}(\tilde{D}_t^c, f_t^\theta)$
17: $\quad\quad$ **else**
18: $\quad\quad\quad \tilde{D}_t^c \leftarrow \tilde{D}_{t-1}^c$
19: $\quad\quad\quad G_t^c \leftarrow \nabla_\theta \big[ \mathcal{L}_{\mathrm{CE}}(\tilde{D}_t^c, \tilde{G}_t^c; f_t^\theta) + \lambda \mathcal{L}_{\mathrm{CON}}(\tilde{D}_t^g, \tilde{D}_t^c; \psi_t^\theta)) \big]$
20: $\quad\quad$ **end if**
21: $\quad\quad$ Uploads $G_t^c$ to Server
22: $\quad$ **end for**
23: $\quad$ **Server:**
24: $\quad G_t^g \leftarrow \mathrm{Aggregate}(G_t^1, ..., G_t^c)$
25: $\quad$ **if** $t \in \tau$ **then**  $\quad\quad\quad\quad\quad\quad\quad\quad\quad\quad\quad\quad$ ▷ Local-global distillation
26: $\quad\quad \tilde{D}_{t+1}^g \leftarrow \mathrm{FederatedGM}(\tilde{D}_t^g, G_t^g)$
27: $\quad\quad$ Send $\tilde{D}_{t+1}^g$ to Clients
28: $\quad$ **end if**
29: $\quad f_{t+1}^\theta \leftarrow \mathrm{ModelUpdate}(G_t^g, f_t^\theta)$
30: $\quad$ Send $f_{t+1}^\theta$ to Clients
31: **end for**

---

a model into two parts, feature extractor $\psi$ and classification head $h$, and the whole classification model is defined as $f^\theta = h \circ \psi$. Given a feature extractor $\psi : \mathbb{R}^d \to \mathbb{R}^{d'}$, we want to generate $\tilde{D}^c$ so that $P_\psi(D^c) \approx P_\psi(\tilde{D}^c)$ where $P_\psi$ is the distribution in feature space. To distill local data during FL efficiently that best fits our task, we intend to use the up-to-date global feature extractor as our kernel function to distill virtual data with global information. Since we can't obtain ground truth distribution of local data, we utilize empirical maximum mean discrepancy (MMD) (Gretton et al., 2012) as our loss function for local virtual distillation:

$$\mathcal{L}_{\mathrm{MMD}} = \sum_k^K \left\lVert \frac{1}{|D_k^c|} \sum_{i=1}^{|D_k^c|} \psi^t(x_i) - \frac{1}{|\tilde{D}_k^{c,t}|} \sum_{j=1}^{|\tilde{D}_k^{c,t}|} \psi^t(\tilde{x}_j^t) \right\rVert^2, \tag{1}$$

where $\psi^t$ and $\tilde{D}^{c,t}$ are the server feature extractor and local virtual data from the latest global iteration $t$. $x_i$ and $\tilde{x}_j^t$ are the data sampled from $D_k^c$ and $\tilde{D}_k^{c,t}$, respectively. $K$ is the total number of classes, and we sum over MMD loss for each class $k \in [K]$. Thus, we obtain updated local virtual data for each FL round.

Although such an efficient distillation strategy is inspired by DM, we highlight the key difference that DM uses randomly initialized model to extract features, whereas we use trained global feature extractor, as the *iterative updating* on the clients' data using the up-to-date network parameters can generate better task-

specific local virtual data. Our intuition comes from the recent success of the empirical neural tangent kernel for data distribution learning and matching (Mohamadi & Sutherland, 2022; Franceschi et al., 2022). Especially, the feature extractor of the model trained with FEDLGD could obtain feature information from other clients, which further harmonizes the domain shift among clients. We apply DM to the baseline FL methods and demonstrate the effectiveness of our proposed iterative strategy in Sec. 5. During distilling global information, FEDLGD only requires a few hundreds steps for, which is computationally efficient.

**Harmonizing local heterogeneity with global anchors.** Data collected in different sites may have different distributions due to different collecting protocols and populations, which degrades the performance of FL. Even more concerning, we find that the issue of data heterogeneity among clients is exacerbated when training with distilled local virtual data in FL (see Fig. 1).To address this, we propose adding a regularization term in the feature space to the total loss function during local model updates, inspired by (Tang et al., 2022).

$$\mathcal{L}_{\text{total}} = \mathcal{L}_{\text{CE}}(\tilde{D}^g, \tilde{D}^c; \theta) + \lambda \mathcal{L}_{\text{Con}}(\tilde{D}^g, \tilde{D}^c; \psi), \tag{2}$$

$$\mathcal{L}_{\text{CE}} = \frac{1}{|\tilde{D}|} \sum_{x,y \in \tilde{D}} -\sum_{k}^{K} y_k log(\hat{y}_k), \hat{y} = f(x; \theta), \tag{3}$$

$$\mathcal{L}_{\text{Con}} = \sum_{j \in B} -\frac{1}{|B_{\backslash j}^{y_j}|} \sum_{x_p \in B_{\backslash j}^{y_j}} \log \frac{e^{(\psi_i(x_j) \cdot \psi_i(x_p)/\tau_{temp})}}{\sum_{x_a \in B_{\backslash j}} e^{(\psi_i(x_j) \cdot \psi_i(x_a)/\tau_{temp})}}. \tag{4}$$

$\mathcal{L}_{\text{CE}}$ is the cross-entropy measured on the virtual data $\tilde{D} = \{\tilde{D}^c, \tilde{D}^g\}$ and K is the number of classes. $\mathcal{L}_{\text{Con}}$ is the supervised contrastive loss (Khosla et al., 2020) for decreasing the feature distances between data from the same class while increasing the feature distances for those from different classes. $B_{\backslash j}$ represents a batch containing both $\tilde{D}^c$ and $\tilde{D}^g$ but without data $j$, $B_{\backslash j}^{y_j}$ is a subset of $B_{\backslash j}$ only with samples belonging to class $y_j$, and $\tau_{temp}$ is a scalar temperature parameter. In such a way, global virtual data can be served for calibration and groups the features of same classes together. At this point, a critical problem arises: *What global virtual data shall we use?*

### 3.3.2 Global Data Distillation for Heterogeneity Harmonization

We claim a 'good' global virtual data should be representative of the global data distributions. Therefore, we propose to leverage local clients' averaged gradients to distill global virtual data, and this process can be naturally incorporated into FL pipeline. We term this global data distillation method as *Federated Gradient Matching*.

**Federated Gradient Matching.** The concept of gradient-based dataset distillation is to minimize the distance between gradients from model parameters trained by original data and virtual data. It is usually considered as a learning-to-learn problem because the procedure consists of model updates and virtual data updates. Zhao *et al.* (Zhao et al., 2021) studies gradient matching in the centralized setting via bi-level optimization that iteratively optimizes the virtual data and model parameters. However, their implementation is not appropriate in our context because there are two fundamental differences in our settings: 1) for model updating, the virtual dataset is on the server side and will not directly optimize the targeted task; 2) for virtual data update, the 'optimal' model comes from the optimized local model aggregation. We argue that these two steps can naturally be embedded in local model updating and global virtual data distillation from the aggregated local gradients. First, we utilize the distance loss $\mathcal{L}_{Dist}$ (Zhao et al., 2021) for gradient matching:

$$\mathcal{L}_{Dist} = Dist(\bigtriangledown_\theta \mathcal{L}_{CE}^{\tilde{D}^g}(\theta), \overline{\bigtriangledown_\theta \mathcal{L}_{CE}^{\tilde{D}^c}}(\theta)), \tag{5}$$

where $\tilde{D}^c$ and $\tilde{D}^g$ denote local and global virtual data, and $\overline{\bigtriangledown_\theta \mathcal{L}_{CE}^{\tilde{D}^c}}$ is the average client gradient. The $Dist(S, T)$ is defined as

$$Dist(S, T) = \sum_{l=1}^{L} \sum_{i=1}^{d_l} (1 - \frac{S_i^l \cdot T_i^l}{||S_i^l|| \, ||T_i^l||}) \tag{6}$$

where $L$ is the number of layers, $S_i^l$ and $T_i^l$ are flattened vectors of gradients corresponding to each output node $i$ from layer $l$, and $d_l$ is the layer output dimension. Then, our proposed federated gradient matching optimize as follows:

$$\min_{D^g} \mathcal{L}_{Dist}(\theta) \quad \text{subject to} \quad \theta = \frac{1}{N} \sum_i^N \theta^{c_i*},$$

where $\theta^{c_i*} = \arg\min_\theta \mathcal{L}_i(\tilde{D}^c)$ is the optimal local model weights of client $i$. By doing federated gradient matching, we gradually distill global virtual data that captures local model information. It is worth noting that we do not need to perform this step for every FL communication round, instead, we find that only selecting a few rounds in the early stage of FL is sufficient to synthesize useful global virtual data, which shares similar insights as reported in (Feng et al., 2023). We provide theoretical analysis to justify the effectiveness of our novel federated gradient matching in lowering the statistical margin in Appendix 4.

## 4   Theoretical Analysis

In this section, we show theoretical insights on FEDLGD. Denote the distribution of global virtual data as $\mathcal{P}_g$ and the distribution of client local virtual data as $\mathcal{P}_c$. In providing theoretical justification for the efficacy of FEDLGD, we can adopt a similar analysis approach as demonstrated in Theorem 3.2 of VHL (Tang et al., 2022), where the relationship between generalization performance and domain misalignment for classification tasks is studied by considering *maximizing* the statistic margin (SM) (Koltchinskii & Panchenko, 2002).

To assess the generalization performance of $f$ with respect to the distribution $\mathcal{P}(x, y)$, we define the SM of FEDLGD as follows:

$$\mathbb{E}_{f=\text{FEDLGD}(\mathcal{P}_g(x,y))} SM_m(f, \mathcal{P}(x, y)), \tag{7}$$

where $m$ is a distance metric, and $f = \text{FEDLGD}(\mathcal{P}_g(x, y))$ means that model $f$ is optimized using FEDLGD with minimizing Eq. 3. Similar to Theorem A.2 of (Tang et al., 2022), we have the lower bound

**Lemma 1 (Lower bound of FedLGD's statistic margin)** *Let $f = \phi \circ \rho$ be a neural network decompose of a feature extractor $\phi$ and a classifier $\rho$. The lower bound of FEDLGD's SM is*

$$\mathbb{E}_{\rho \leftarrow \mathcal{P}_g} SM_m(\rho, \mathcal{P}) \geq \mathbb{E}_{\rho \leftarrow \mathcal{P}_g} SM_m(\rho, \tilde{D}) - \left| \mathbb{E}_{\rho \leftarrow \mathcal{P}_g} \left[ SM_m(\rho, \mathcal{P}_g) - SM_m(\rho, \tilde{D}) \right] \right| - \mathbb{E}_y d \left( \mathcal{P}_c(\phi \mid y), \mathcal{P}_g(\phi \mid y) \right).$$

**Proof 1** *Following proof in Theorem A.2 of (Tang et al., 2022), the statistical margin is decomposed as*

$$\mathbb{E}_{\rho \leftarrow \mathcal{P}_g} SM_m(\rho, \mathcal{P})$$
$$\geq \mathbb{E}_{\rho \leftarrow \mathcal{P}_g} SM_m(\rho, \tilde{D}) - \left| \mathbb{E}_{\rho \leftarrow \mathcal{P}_g} \left[ SM_m(\rho, \mathcal{P}_g) - SM_m(\rho, \tilde{D}) \right] \right| - \left| \mathbb{E}_{\rho \leftarrow \mathcal{P}_g} \left[ SM_m(\rho, \mathcal{P}) - SM_m(\rho, \mathcal{P}_g) \right] \right|$$
$$\geq \mathbb{E}_{\rho \leftarrow \mathcal{P}_g} SM_m(\rho, \tilde{D}) - \left| \mathbb{E}_{\rho \leftarrow \mathcal{P}_g} \left[ SM_m(\rho, \mathcal{P}_g) - SM_m(\rho, \tilde{D}) \right] \right| - \mathbb{E}_y d \left( \mathcal{P}(\phi \mid y), \mathcal{P}_g(\phi \mid y) \right)$$

Another component in our analysis is building the connection between our used gradient matching strategy and the distribution match term in the bound.

**Lemma 2 (Proposition 2 of (Yu et al., 2023))** *First-order distribution matching objective is approximately equal to gradient matching of each class for kernel ridge regression models following a random feature extractor.*

**Theorem 1** *Due to the complexity of data distillation steps, without loss of generality, we consider kernel ridge regression models with a random feature extractor. Minimizing total loss of FEDLGD (Eq. 2) for harmonizing local heterogeneity with global anchors elicits a model with bounded statistic margin (i.e.,the upper bound of the SM bound in Theorem 1).*

**Proof 2** *The first and second term can be bounded by maximizing SM of local virtual training data and global virtual data. The large SM of global virtual data distribution $\mathcal{P}_g(x, y)$ is encouraged by minimizing cross-entropy $L_{CE}(\tilde{D}^g, y)$ in our objective function Eq. 3.*

Table 1: Test accuracy for `DIGITS` under different images per class (IPC) and model architectures. R and C stand for ResNet18 and ConvNet, respectively, and we set IPC to 10 and 50. 'Average' is the unweighted test accuracy average of all the clients. The best results are marked in **bold**.

| DIGITS | | MNIST | | SVHN | | USPS | | SynthDigits | | MNIST-M | | Average | |
|---|---|---|---|---|---|---|---|---|---|---|---|---|---|---|
| IPC | | 10 | 50 | 10 | 50 | 10 | 50 | 10 | 50 | 10 | 50 | 10 | 50 |
| FedAvg | R | 73.0 | 92.5 | 20.5 | 48.9 | 83.0 | 89.7 | 13.6 | 28.0 | 37.8 | 72.3 | 45.6 | 66.3 |
| | C | 94.0 | 96.1 | 65.9 | 71.7 | 91.0 | 92.9 | 55.5 | 69.1 | 73.2 | 83.3 | 75.9 | 82.6 |
| FedProx | R | 72.6 | 92.5 | 19.7 | 48.4 | 81.5 | 90.1 | 13.2 | 27.9 | 37.3 | 67.9 | 44.8 | 65.3 |
| | C | 93.9 | 96.1 | 66.0 | 71.5 | 90.9 | 92.9 | 55.4 | 69.0 | 73.7 | 83.3 | 76.0 | 82.5 |
| FedNova | R | 75.5 | 92.3 | 17.3 | 50.6 | 80.3 | 90.1 | 11.4 | 30.5 | 38.3 | 67.9 | 44.6 | 66.3 |
| | C | 94.2 | 96.2 | 65.5 | 73.1 | 90.6 | 93.0 | 56.2 | 69.1 | 74.6 | 83.7 | 76.2 | 83.0 |
| Scaffold | R | 75.8 | 93.4 | 16.4 | 53.8 | 79.3 | 91.3 | 11.2 | 34.2 | 38.3 | 70.8 | 44.2 | 68.7 |
| | C | 94.1 | 96.3 | 64.9 | 73.3 | 90.6 | 93.4 | 56.0 | 70.1 | 74.6 | 84.7 | 76.0 | 83.6 |
| MOON | R | 15.5 | 80.4 | 15.9 | 14.2 | 25.0 | 82.4 | 10.0 | 11.5 | 11.0 | 35.4 | 15.5 | 44.8 |
| | C | 85.0 | 95.5 | 49.2 | 70.5 | 83.4 | 92.0 | 31.5 | 67.2 | 56.9 | 82.3 | 61.2 | 81.5 |
| FedProto | R | 13.5 | 56.7 | 9.3 | 7.8 | 39.6 | 79.7 | 10.0 | 10.6 | 10.0 | 11.2 | 16.5 | 33.2 |
| | C | 91.9 | 96.8 | 52.7 | 73.9 | 93.3 | 96.6 | 27.2 | 52.8 | 69.0 | 84.3 | 668 | 80.9 |
| VHL | R | 87.8 | 95.9 | 29.5 | 67.0 | 88.0 | 93.5 | 18.2 | 60.7 | 52.2 | **85.7** | 55.1 | 80.5 |
| | C | 95.0 | 96.9 | **68.6** | 75.2 | 92.2 | 94.4 | 60.7 | 72.3 | 76.1 | 83.7 | 78.5 | 84.5 |
| FEDLGD | R | **92.9** | **96.7** | 46.9 | **73.3** | **89.1** | **93.9** | **27.9** | **72.9** | **70.8** | 85.2 | **65.5** | **84.4** |
| | C | **95.8** | **97.1** | 68.2 | **77.3** | **92.4** | **94.6** | **67.4** | **78.5** | **79.4** | **86.1** | **80.6** | **86.7** |

*The third term represents the discrepancy of distributions of virtual and real data. We denote this term as $\mathcal{D}^{\mathcal{P}_c}_{\phi|y}(\mathcal{P}_g) = \mathbb{E}_y d\left(\mathcal{P}_c(\phi \mid y), \mathcal{P}_g(\phi \mid y)\right)$ and aim to show that $\mathcal{D}^{\mathcal{P}_c}_{\phi|y}(\mathcal{P}_g)$ can achieve small upper bound under proper assumptions.*

*Based on Lemma 2, the first-order distribution matching objective $\mathcal{D}^{\mathcal{P}_c}_{\phi|y}(\mathcal{P}_g)$ is approximately equal to gradient matching of each class, as shown in objective $\mathcal{L}_{Dist}$ (Eq. 5). Namely, minimizing gradient matching objective $\mathcal{L}_{Dist}$ in FEDLGD implies minimizing $\mathcal{D}^{\mathcal{P}_c}_{\phi|y}(\mathcal{P}_g)$ in the setting. Hence, using gradient matching generated global virtual data elicits the model's SM a tight lower bound.*

**Remark 1** *The key distinction between FEDLGD and VHL primarily lies in the final term, which is exactly a distribution matching objective. It is important to note that in VHL, the global virtual data is generated from an un-pretrained StyleGAN, originating from various Gaussian distributions, which we denote as $\mathcal{P}_g$. The VHL paper only provided a lower bound for $\mathcal{D}^{\mathcal{P}_c}_{\phi|y}(\mathcal{P}_g)$ but did not show how it is upper bounded. However, for the purpose of maximizing SM to achieve strong generalization, we want to show SM has a tight lower bound. Therefore, upper bounded the last term is desired. In contrast, our approach employs the gradient matching strategy to synthesize the global virtual data. To prove our performance improvement, we can show that FEDLGD could achieve a tight lower bound for SM.*

## 5 Experiment

To evaluate FEDLGD, we consider the FL setting in which clients obtain data from different domains with the same target task. Specifically, we compare with multiple baselines on **benchmark datasets** `DIGITS`, where each client has data from completely different open-sourced datasets. The experiment aims to show that FEDLGD can effectively mitigate large domain shifts. Additionally, we evaluate the performance of FEDLGD on another **large benchmark dataset**, `CIFAR10C` (Hendrycks & Dietterich, 2019), which collects data with different corruptions yielding data distribution shift and contains a large number of clients, so that we can investigate varied client sampling in FL. The experiment aims to show FEDLGD's feasibility on large-scale FL environments. We also validate the performance under **real medical datasets**, `RETINA`.

Table 2: Averaged test accuracy for `CIFAR10C` with ConvNet.

| CIFAR10C | | FedAvg | | FedProx | | FedNova | | Scaffold | | MOON | | FedProto | | VHL | | FEDLGD | |
|---|---|---|---|---|---|---|---|---|---|---|---|---|---|---|---|---|---|---|
| IPC | | 10 | 50 | 10 | 50 | 10 | 50 | 10 | 50 | 10 | 50 | 10 | 50 | 10 | 50 | 10 | 50 |
| | 0.2 | 27.0 | 44.9 | 27.0 | 44.9 | 26.7 | 34.1 | 27.0 | 44.9 | 20.5 | 31.3 | 14.4 | 26.3 | 21.8 | 45.0 | **32.9** | **46.8** |
| Client ratio | 0.5 | 29.8 | 51.4 | 29.8 | 51.4 | 29.6 | 45.9 | 30.6 | 51.6 | 23.8 | 43.2 | 16.4 | 36.1 | 29.3 | 51.7 | **39.5** | **52.8** |
| | 1 | 33.0 | 54.9 | 33.0 | 54.9 | 30.0 | 53.2 | 33.8 | 54.5 | 26.4 | 51.6 | 20.3 | 40.4 | 34.4 | 55.2 | **47.6** | **57.4** |

## 5.1 Training and Evaluation Setup

**Model architecture.** We adapt ResNet18 (He et al., 2016) and ConvNet (Zhao et al., 2021) (detailed in Appendix C.4) in our study. To achieve the optimal performance, we apply the same architecture to perform both the local distillation task and the classification task, as suggested in (Zhao et al., 2021).

**Comparison methods.** We compare the performance of downstream classification tasks using state-of-the-art heterogeneous FL algorithms, FedAvg (McMahan et al., 2017), FedProx (Li et al., 2020), FedNova (Wang et al., 2020), Scaffold (Karimireddy et al., 2020), MOON (Li et al., 2021), FedProto (Tan et al., 2022), and VHL (Tang et al., 2022). We use local virtual data from our initialization stage for FL methods other than ours and perform classification on client's testing set and report the test accuracies.

**FL training setup.** We use the SGD optimizer to update local models. If not specified, our default setting for learning rate is $10^{-2}$, local model update epochs is 1, total update rounds is 100, the batch size for local training is 32, and the number of virtual data update iterations ($|\tau|$) is 10. The numbers of default virtual data distillation steps for clients and server are set to 100 and 500, respectively. Since we only have a few clients for `DIGITS`, we will select all the clients for each iteration, while the client selection criteria for `CIFAR10C` experiments will be specified in Sec. 5.3.

**Proper Initialization for Distillation.** For privacy concerns and model performance, we initialize local virtual data using local statistics for local data distillation. Specifically, each client calculates the statistics of its own data for each class, denoted as $\mu_i^c, \sigma_i^c$, and then initializes the distillation images per class, $x \sim \mathcal{N}(\mu_i^c, \sigma_i^c)$, where $c$ and $i$ represent each client and categorical label. For privacy consideration, we use random noise as initialization for global virtual data distillation. The comparison between different initialization strategies can be found in Appendix B.

## 5.2 `DIGITS` Experiment

**Datasets.** We use the following datasets for our benchmark experiments: `DIGITS` = {MNIST (LeCun et al., 1998), SVHN (Netzer et al., 2011), USPS (Hull, 1994), SynthDigits (Ganin & Lempitsky, 2015), MNIST-M (Ganin & Lempitsky, 2015)}. Each dataset in `DIGITS` contains handwritten, real street and synthetic digit images of $0, 1, \cdots, 9$. As a result, we have 5 clients in the experiments.

**Comparison under various conditions.** To validate the effectiveness of FEDLGD, we first compare it with the alternative FL methods varying on two important factors: Image-per-class (IPC) and different deep neural network architectures (arch). We use IPC $\in \{10, 50\}$ and arch $\in$ { ResNet18(R), ConvNet(C)} to examine the performance of SOTA models and FEDLGD using distilled `DIGITS`. Note that we fix IPC = 10 for global virtual data and vary IPC for local virtual data. Tab. 1 shows the test accuracies of `DIGITS` experiments. One can observe that for each FL algorithm, ConvNet(C) always has the best performance under all IPCs. The observation is consistent with (Zhao & Bilen, 2023) as more complex architectures may cause over-fitting to virtual data. It is also shown that using IPC = 50 always outperforms IPC = 10 as expected since more virtual data can captures more real data distribution and thus facilitates model training. Overall, FEDLGD outperforms other SOTA methods, where on average accuracy, FEDLGD increases the best test accuracy results among the baseline methods of 2.1% (IPC =10, arch = C), 10.4% (IPC =10, arch = R), 2.2% (IPC = 50, arch = C) and 3.9% (IPC =50, arch = R). VHL is the closest strategy to FEDLGD and achieves the best performance among the baseline methods, indicating that the feature alignment solutions are promising for handling heterogeneity in federated virtual learning. However, the worse performance may result from the differences in synthesizing global virtual data. VHL uses untrained StyleGAN (Karras et al., 2019) to generate global virtual data without further updating. On the contrary, we gradually update our global virtual data during FL training.

### 5.3 `CIFAR10C` Experiment

**Datasets.** We conduct large-scale FL experiments on `CIFAR10C`[1], where, like previous studies (Li et al., 2021), we apply Dirichlet distribution with $\alpha = 2$ to generate 3 partitions on each distorted Cifar10-C (Hendrycks & Dietterich, 2019), resulting in 57 *domain and label heterogeneous* non-IID clients. In addition, we randomly sample a fraction of clients with ratio = 0.2, 0.5, and 1 for each FL round.

**Comparison under different client sampling ratios.** The objective of the experiment is to test FEDLGD under popular FL questions: class imbalance, large number of clients, different client sample ratios, and domain and label heterogeneity. One benefit of federated virtual learning is that we can easily handle class imbalance by distilling the same number (IPC) of virtual data. We will vary IPC and fix the model architecture to ConvNet since it is validated to yield better performance in virtual training. One can observe from Tab. 2 that FEDLGD consistently achieves the best performance under different IPC and client sampling ratios. We would like to point out that when IPC=10, the performance boosts are significant, which indicates that FEDLGD is well-suited for FL when there is a large group of clients with limited number of local virtual data.

### 5.4 `RETINA` Experiment

**Dataset.** For medical dataset, we use the retina image datasets, `RETINA` = {Drishti (Sivaswamy et al., 2014), Acrima (Diaz-Pinto et al., 2019), Rim (Batista et al., 2020), Refuge (Orlando et al., 2020)}, where each dataset contains retina images from different stations with image size $96 \times 96$, thus forming four clients in FL. We perform binary classification to identify *Glaucomatous* and *Normal*. Example images and distributions can be found in Appendix C.3.

**Comparison with baselines.** The results for `RETINA` experiments are shown in Table 3, where D, A, Ri, Re represent Drishti, Acrima, Rim, and Refuge datasets. We only set IPC=10 for this experiment as clients in `RETINA` contain much fewer data points. The learning rate is set to $10^{-3}$. FEDLGD has the best performance compared to the other methods w.r.t the unweighted averaged accuracy (Avg) among clients. To be precise, FEDLGD increases unweighted averaged test accuracy for 3.1%(versus the best baseline) on ConvNet. The same accuracy for different methods is due to the limited number of testing samples. We conjecture the reason why VHL (Tang et al., 2022) has lower performance improvement in `RETINA` experiments is that this dataset is in higher dimensional and clinical diagnosis evidence on fine-grained details, *e.g.*,

Table 3: Test accuracy for `RETINA` experiments under different model architectures and IPC=10. We have 4 clients: Drishti(D), Acrima(A), Rim(Ri), and Refuge(Re), respectively. We also show the average test accuracy (Avg). The same accuracy for different methods is due to the limited number of testing samples.

| RETINA | | D | A | Ri | Re | Avg |
|---|---|---|---|---|---|---|
| FedAvg | C | 69.4 | 84.0 | **88.0** | 86.5 | 82.0 |
| FedProx | C | 68.4 | 84.0 | **88.0** | 86.5 | 81.7 |
| FedNova | C | 68.4 | 84.0 | **88.0** | 86.5 | 81.7 |
| Scaffold | C | 68.4 | 84.0 | **88.0** | 86.5 | 81.7 |
| MOON | C | 57.9 | 72.0 | 76.0 | 85.0 | 72.7 |
| FedProto | C | 73.6 | 86.0 | 54.0 | 77.5 | 72.8 |
| VHL | C | 68.4 | 78.0 | 81.0 | 87.0 | 78.6 |
| FEDLGD | C | **78.9** | **86.0** | **88.0** | **87.5** | **85.1** |

cup-to-disc ratio and disc rim integrity (Schuster et al., 2020). Therefore, it is difficult for untrained Style-GAN (Karras et al., 2019) to serve as anchor for this kind of larger images.

### 5.5 Ablation studies for FedLGD

The success of FEDLGD relies on the novel design of local-global data distillation, where the selection of regularization loss and the number of iterations for data distillation play a key role. Recall that among the total FL training epochs, we perform local-global distillation on the selected $\tau$ *iterations*, where the server and clients will perform data updating for some pre-defined *steps*. Thus, we will study the choice of regularization loss and its weighting ($\lambda$) in the total loss function, as well as the effect of *iterations* and *steps*. By default, we use ConvNet, global IPC=10, local IPC=50, $|\tau|$=10, and (local, global) update *steps*=(100,

---

[1]Cifar10-C is a collection of augmented Cifar10 that applies 19 different corruptions, resulting in $6k \times 19 = 114k$ data points.

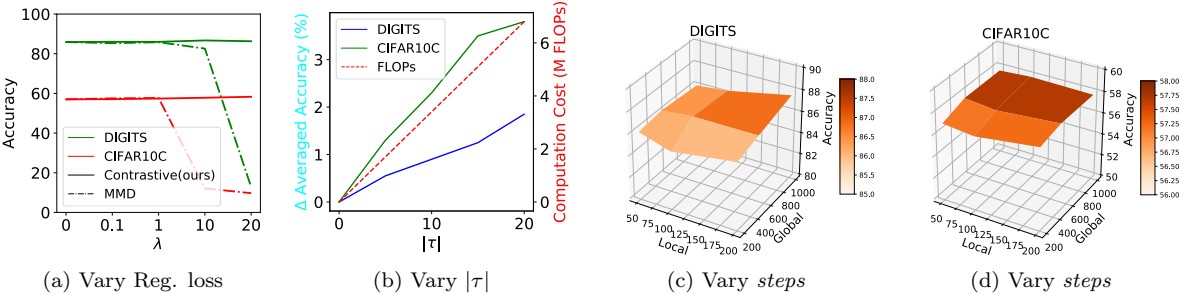

(a) Vary Reg. loss      (b) Vary $|\tau|$      (c) Vary *steps*      (d) Vary *steps*

Figure 3: (a) Comparison between different regularization losses and their weightings($\lambda$). One can observe that $\mathcal{L}_{\mathrm{Con}}$ gives us better and more stable performance with different coefficient choices. (b) The solid curves describes the improved accuracy compared to $|\tau| = 0$, and the dashed curve indicates the computation cost. The model performance improves with the increasing $|\tau|$, which is a trade-off between computation cost and model performance. Vary data updating *steps* for (c) DIGITS and (d) CIFAR10C. FEDLGD yields consistent performance, and the accuracy improves with an increasing number of local and global steps.

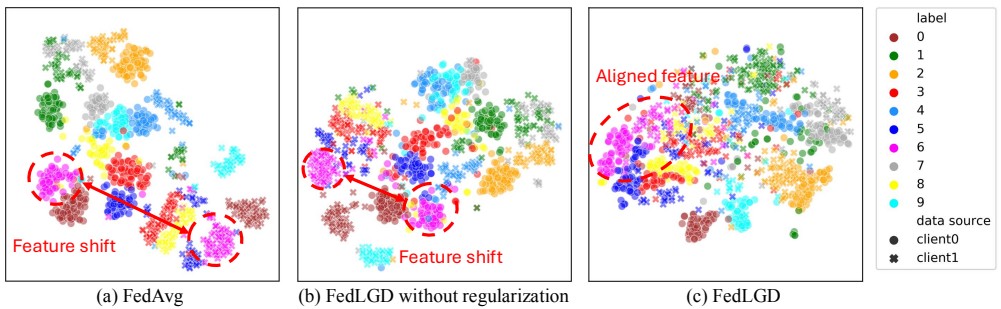

(a) FedAvg      (b) FedLGD without regularization      (c) FedLGD

Figure 4: tSNE plots on feature space for FedAvg, FEDLGD without regularization, and FEDLGD.

500). We also discuss computation cost and privacy, two important factors in FL. Further ablation studies can be found in Appendix B.

**Effect of regularization loss.** FEDLGD uses supervised contrastive loss $\mathcal{L}_{\mathrm{Con}}$ as a regularization term to encourage local and global virtual data embedding into a similar feature space. To demonstrate its effectiveness, we perform ablation studies to replace $\mathcal{L}_{\mathrm{Con}}$ with an alternative distribution similarity measurement, MMD loss, with different $\lambda$'s ranging from 0 to 20. Fig. 3a shows the average test accuracy. Using $\mathcal{L}_{\mathrm{Con}}$ gives us better and more stable performance with different $\lambda$ choices. We select $\lambda$=10 and 1 for DIGITS and CIFAR10C, respectively. It is worth noting that when $\lambda = 0$, FEDLGD can still yield competitive accuracy, which indicates the utility of our local and global virtual data. To explain the effect of our proposed regularization loss on feature representations, we embed the latent features before fully-connected layers to a 2D space using tSNE (Van der Maaten & Hinton, 2008) shown in Fig. 4. For the model trained with FedAvg (Fig. 4a), features from two clients ($\times$ and $\circ$) are closer to their own distribution regardless of the labels (colors). In Fig. 4b, we perform virtual FL training but without the regularization term (Eq. 4). Fig. 4c shows FEDLGD, and one can observe that data from different clients with the same label are grouped together.

**Analysis of distillation *iterations* ($|\tau|$).** Fig. 3b shows the improved averaged test accuracy if we increase the number of distillation iterations with FEDLGD. The base accuracy for DIGITS and CIFAR10C are 85.8 and 55.2 when $\tau = \emptyset$. We fix local and global update *steps* to 100 and 500, and the selected iterations ($\tau$) are defined as arithmetic sequences with $d = 5$ (i.e., $\tau = \{0, 5, ...\}$). One can observe that the model performance improves with the increasing $|\tau|$. This is because we obtain better virtual data with more local-global distillation iterations, which is a trade-off between computation cost and model performance.

**Robustness on virtual data update *steps*.** In Fig. 3c and Fig. 3d, we vary (local, global) data updating steps. One can observe that FEDLGD yields stable performance (always outperforms baselines), and the accuracy slightly improves with an increasing number of local and global steps.

**Computation cost.** We have shown the increased computation cost caused by increasing the number of selected rounds $|\tau|$ in Fig. 3b. Here, we discuss the overall accumulated computation cost for the 100 total FL training rounds, including both selected and unselected iterations in Fig. 5. The computation costs for FEDLGD in DIGITS and CIFAR10C are identical since we use IPC=50 for training. For RETINA, since we apply IPC=10, FEDLGD has significant efficiency improvement. Overall, FEDLGD reduces the computation cost on the clients' side by training with virtual data compared to classical FedAvg that train on real datasets.

**Privacy.** We note that FEDLGD uses *pre-existing* information, *i.e.*, shared averaged gradients and global model, to distill virtual data, so there is no extra privacy leakage. Like the standard FL training, FEDLGD may be vulnerable to deep privacy attacks, such as membership inference attacks (MIAs) (Shokri et al., 2017) and gradient inversion attacks(GIAs) (Zhu et al., 2019; Huang et al., 2021).We empirically show FEDLGD can potentially defend both attacks, which is also implied by (Xiong et al., 2023; Dong et al., 2022). Preserv-

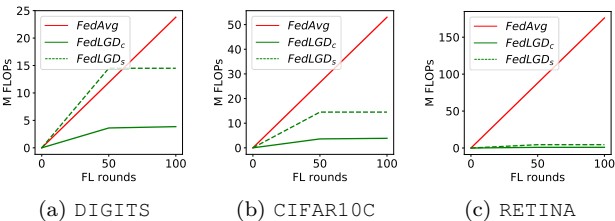

Figure 5: FEDLGD reduces the Accumulated computation cost on the clients' side compared to FedAvg.

ing identity-level privacy can be further improved by employing differential privacy (Abadi et al., 2016) in dataset distillation, such as applying DPSGD during local data distillation or applying DPSGD on the local gradients, but this goes beyond the main focus of our work.

MIAs (Shokri et al., 2017) aims to identify if a given data point belongs to the model's training data. We compare the performance of MIA directly on models trained with original data (FedAvg) and with the synthetic dataset (FEDLGD). If the MIA performance on the original images is worse than the one on FedAvg, we claim that the synthetic data helps with privacy. Here, we implemented the Likelihood Ratio MIA (Carlini et al., 2022a), where the gradients are collected for the server model on training and testing data individually. The likelihood of the point belonging to the training set is then obtained using the Gaussian kernel density estimation (Fig. 6). If the ROC curve intersects with the diagonal dashed line (representing a random membership

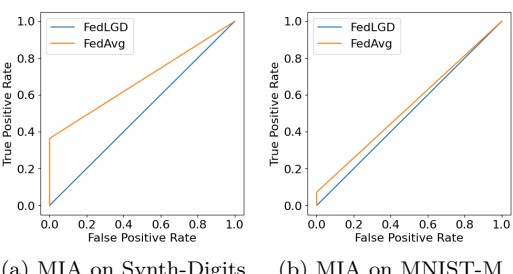

Figure 6: MIA results on models trained with FedAvg (using original dataset) and FEDLGD (using distilled virtual dataset). If the ROC curve is the same as the diagonal line, it means the membership cannot be inferred.

classifier), it signifies that the approach provides a stronger defense against membership inference compared to the method with a larger area under the ROC curve. FEDLGD results in ROC curves that are more closely aligned with the diagonal line, suggesting that attacking membership becomes more challenging.

Using dataset distillation to synthesize virtual data can be shown to mitigate against gradient-based inversion attacks (GIAs) (Geiping et al., 2020; Huang et al., 2021). Here, we use Cifar10 (Krizhevsky et al., 2009) as an example. We perform local training on a ConvNet from one client in CIFAR10Cand apply gradient inversion attack to reconstruct the raw images. Then, we evaluate the reconstruction quality using perceptual loss (LPIPS) (Zhang et al., 2018). As a result, the reconstructed distilled image is visually different from raw images, and it effectively alleviates the attack from perceptual perspective, by reducing LPIPS from 0.253 to 0.177. Note that

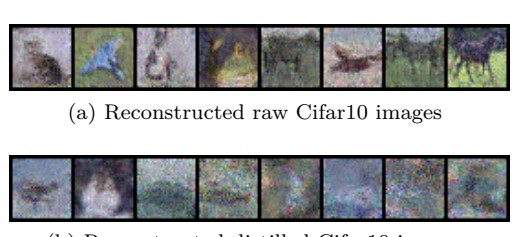

(a) Reconstructed raw Cifar10 images

(b) Reconstructed distilled Cifar10 images

Figure 7: GIA on raw and distilled Cifar10 images.

in FEDLGD, the shared global virtual data is synthesized by the *averaged* gradients, which further improves the privacy guarantee.

## 6 Conclusion

In this paper, we introduce a new approach for FL, called FEDLGD. It utilizes virtual data on both client and server sides to train FL models. We are the first to reveal that FL on distilled local virtual data can increase heterogeneity. To tackle the heterogeneity issue, we seamlessly integrated dataset distillation algorithms into FL pipeline by proposing iterative distribution matching and federated gradient matching to iteratively update local and global virtual data. Then, we apply global virtual regularization to effectively harmonize domain shift. Our experiments on benchmark and real medical datasets show that FEDLGD outperforms current state-of-the-art methods in heterogeneous settings. Furthermore, FEDLGD can be combined with other model-synchronization-based FL approaches to further improve its performance. The potential limitation lies in the additional communication and computation cost in data distillation, but we show that the trade-off is acceptable and can be mitigated by decreasing distillation *iterations* and *steps*. Our future direction includes investigating privacy-preserving data generation and utilizing the synthesized global virtual data for federated continual learning or training personalized models. We believe that this work sheds light on how to effectively mitigate data heterogeneity from a dataset distillation perspective and will inspire future work to enhance FL performance, privacy, and efficiency.

## Acknowledgement

This work is supported in part by the Natural Sciences and Engineering Research Council of Canada (NSERC), CIFAR AI Chair Awards, CIFAR AI Catalyst Grant, NVIDIA Hardware Award, UBC Sockeye, and Compute Canada Research Platform.

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
