**Road Map of Appendix** Our appendix is organized into six sections. The notation table is in Appendix A, which contains the mathematical notations for Algorithm 1, which outlines the pipeline of FEDLGD. Appendix B provides a list of ablation studies to analyze FEDLGD, including communication overhead, convergence rate under different random seeds, and hyper-parameter choices. Last but not least, Appendix C lists the details of our experiments, including the data set information and model architectutres. Our code and model checkpoints are available in this https://github.com/ubc-tea/FedLGD.

## A   Notation Table

| Notations | Description |
|:---:|:---|
| $d$ | input dimension |
| $d'$ | feature dimension |
| $f^\theta$ | global model |
| $\theta$ | model parameters |
| $\psi$ | feature extractor |
| $h$ | projection head |
| $D^g, D^c$ | original global and local data |
| $\tilde{D}^g, \tilde{D}^c$ | global and local synthetic data |
| $\mathcal{L}_{\text{total}}$ | total loss function for virtual federated training |
| $\mathcal{L}_{\text{CE}}$ | cross-entropy loss |
| $\mathcal{L}_{\text{Dist}}$ | Distance loss for gradient matching |
| $\mathcal{L}_{\text{MMD}}$ | MMD loss for distribution matching |
| $\mathcal{L}_{\text{Con}}$ | Contrastive loss for local training regularization |
| $\lambda$ | coefficient for local training regularization term |
| $T$ | total training iterations |
| $\tau$ | selected local global distillation iterations |

Table 4: Important notations used in the paper.

## B   Additional Results and Ablation Studies for FedLGD

### B.1   Communication overhead.

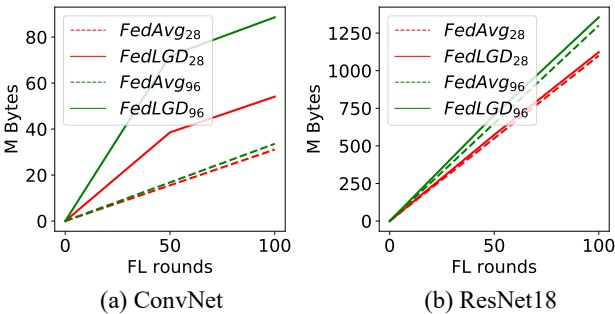

(a) ConvNet                    (b) ResNet18

Figure 8: Accumulated communication overhead compared to classical FedAvg.

The accumulated communication overhead for image size $28 \times 28$ and $96 \times 96$ can be found in Fig. 8. We show the communication cost for both ConvNet and ResNet18. Note that the trade-off of our design reflects in the increased communication overhead, where the clients need to *download* the latest global virtual data in the selected rounds ($\tau$). However, we argue that the $|\tau|$ can be adjusted based on the communication budget. Additionally, as the model architecture becomes more complex, the added communication overhead

turns out to be minor. For instance, the difference between the dashed and solid lines in Fig. 8(b) is less significant than the difference observed in Fig. 8(a).

## B.2 Different random seeds

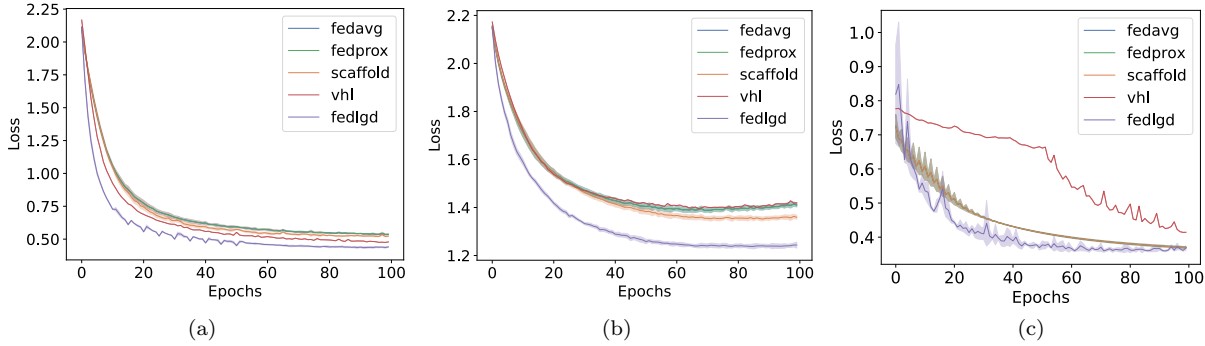

Figure 9: Averaged testing loss for (a) `DIGITS` with IPC = 50, (b) `CIFAR10C` with IPC = 50, and (c) `RETINA` with IPC = 10 experiments.

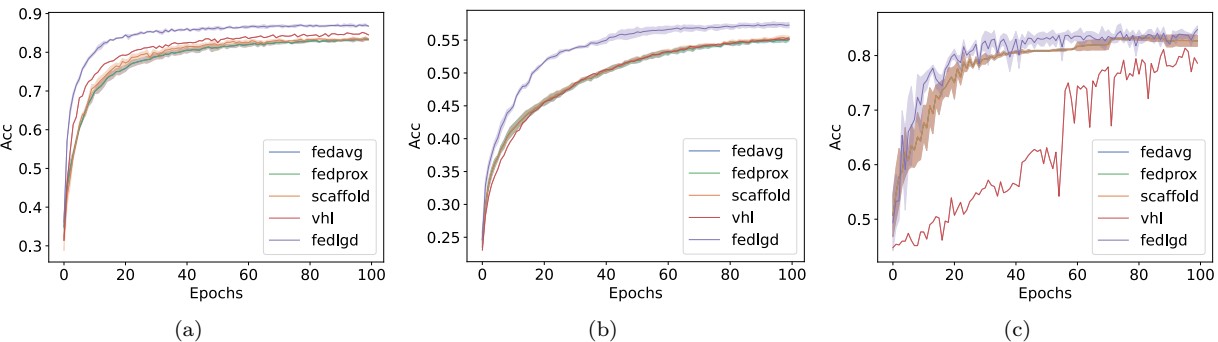

Figure 10: Averaged testing accuracy for (a) `DIGITS` with IPC = 50, (b) `CIFAR10C` with IPC = 50, and (c) `RETINA` with IPC = 10 experiments.

To show the consistent performance of FEDLGD, we repeat the experiments for `DIGITS`, `CIFAR10C`, and `RETINA` with three random seeds, and report the validation loss and accuracy curves in Figure 9 and 10 (The standard deviations of the curves are plotted as shadows.). We use ConvNet for all the experiments. IPC is set to 50 for `CIFAR10C` and `DIGITS`; 10 for `RETINA`. We use the default hyperparameters for each dataset, and only report FedAvg, FedProx, Scaffold, VHL, which achieves the best performance among baseline as indicated in Table 1, 2, and 3 for clear visualization. One can observe that FEDLGD has faster convergence rate and results in optimal performances compared to other baseline methods.

## B.3 Different heterogeneity levels of label shift

In the experiment presented in Sec 5.3, we study FEDLGD under both label and domain shifts, where labels are sampled from Dirichlet distribution. To ensure dataset distillation performance, we ensure that each class at least has 100 samples per client, thus setting the coefficient of Dirichlet distribution $\alpha = 2$ to simulate the worst case of label heterogeneity that meets the quality dataset distillation requirement. Here, we show the performance with a less heterogeneity level ($\alpha = 5$) while keeping the other settings the same as those in Sec.5.3. The results are shown in Table 5. As we expect, the performance drop when the heterogeneity level increases ($\alpha$ decreases). One can observe that when heterogeneity increases, FEDLGD's performance drop

less except for VHL. We conjecture that VHL yields similar test accuracy for $\alpha = 2$ and $\alpha = 5$ is that it uses fixed global virtual data so that the effectiveness of regularization loss does not improve much even if the heterogeneity level is decreased. Nevertheless, FEDLGD consistently outperforms all the baseline methods.

Table 5: Comparison of different $\alpha$ for Drichilet distribution on CIFAR10C.

| alpha | 2 | 5 |
|---|---|---|
| FedAvg | 54.9 | 55.4 |
| FedProx | 54.9 | 55.4 |
| FedNova | 53.2 | 55.4 |
| Scaffold | 54.5 | 55.6 |
| MOON | 51.6 | 51.1 |
| VHL | 55.2 | 55.4 |
| FedLGD | 57.4 | 58.1 |

## B.4    Analysis of batch size

Batch size is another factor for training the FL model and our distilled data. We vary the batch size $\in \{8, 16, 32, 64\}$ to train models for CIFAR10C with the fixed default learning rate. We show the effect of batch size in Table 6 reported on average testing accuracy. One can observe that the performance is slightly better with moderately smaller batch size which might due to two reasons: 1) more frequent model update locally; and 2) larger model update provides larger gradients, and FEDLGD can benefit from the large gradients to distill higher quality virtual data. Overall, the results are generally stable with different batch size choices.

Table 6: Varying batch size in FEDLGD on CIFAR10C. We report the unweighted accuracy. One can observe that the performance increases when the batch size decreases.

| Batch Size | 8 | 16 | 32 | 64 |
|---|---|---|---|---|
| CIFAR10C | 59.5 | 58.3 | 57.4 | 56.0 |

## B.5    Analysis of Local Epoch

Aggregating at different frequencies is known as an important factor that affects FL behavior. Here, we vary the local epoch $\in \{1, 2, 5\}$ to train all baseline models on CIFAR10C. Figure 11 shows the result of test accuracy under different epochs. One can observe that as the local epoch increases, the performance of FEDLGD would drop a little bit. This is because doing gradient matching requires the model to be trained to an intermediate level, and if local epochs increase, the loss of CIFAR10C models will drop significantly. However, FEDLGD still consistently outperforms the baseline methods. As our future work, we will investigate the tuning of the learning rate in the early training stage to alleviate the effect.

## B.6    Different Initialization for Virtual Images

To validate our proposed initialization for virtual images has the best trade-off between privacy and efficacy, we compare our test accuracy with the models trained with synthetic images initialized by random noise and real images in Table 7. To show the effect of initialization under large domain shift, we run experiments on DIGITS dataset. One can observe that our method which utilizes the statistics $(\mu_i, \sigma_i)$ of local clients as initialization outperforms random noise initialization. Although our performance is slightly worse than the initialization that uses real images from clients, we do not ask the clients to share real image-level information to the server which is more privacy-preserving.

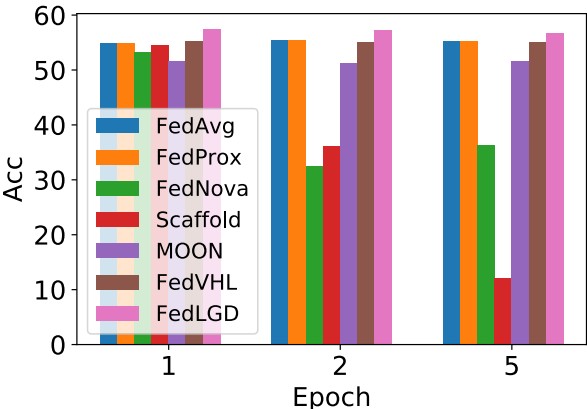

Figure 11: Comparison of model performances under different local epochs with `CIFAR10C`.

Table 7: Comparison of different initialization for synthetic images `DIGITS`. Ours ($\mathcal{N}(\mu_i, \sigma_i)$) is shown in the middle column.

| DIGITS | $\mathcal{N}(0,1)$ | $\mathcal{N}(\mu_i, \sigma_i)$ | Real Images |
|---|---|---|---|
| MNIST | 96.3 | 97.1 | 97.7 |
| SVHN | 75.9 | 77.3 | 78.8 |
| USPS | 933 | 94.6 | 94.2 |
| SynthDigits | 72.0 | 78.5 | 82.4 |
| MNIST-M | 83.7 | 86.1 | 89.5 |
| Average | 84.2 | 86.7 | 88.5 |

## C  Experimental details

### C.1  Visualization of the original images

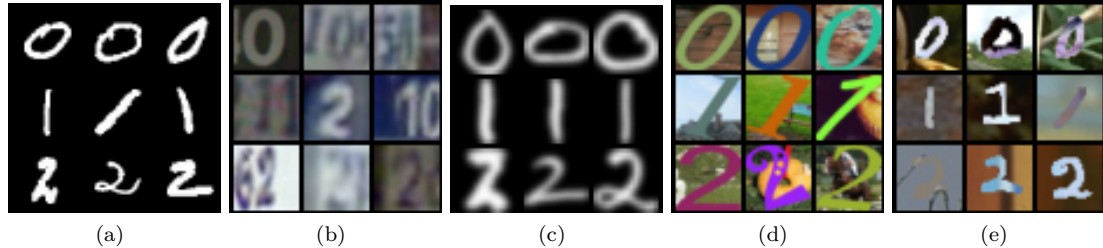

(a)    (b)    (c)    (d)    (e)

Figure 12: Visualization of the original digits dataset. (a) visualized the MNIST client; (b) visualized the SVHN client; (c) visualized the USPS client; (d) visualized the SynthDigits client; (e) visualized the MNIST-M client.

The visualization of the original `DIGITS`, `CIFAR10C`, and `RETINA` images can be found in Figure 12, Figure 13, and Figure 14, respectively.

### C.2  Visualization of our distilled global and local images

The visualization of the virtual `DIGITS`, `CIFAR10C`, and `RETINA` images can be found in Figure 15, Figure 16, and Figure 17, respectively.

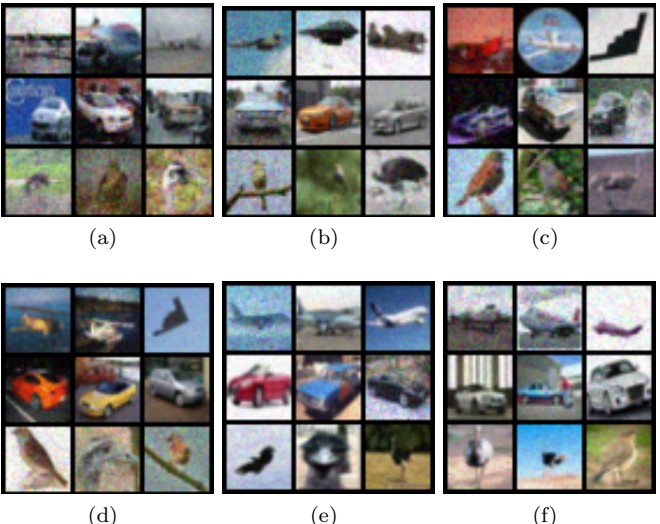

Figure 13: Visualization of the original `CIFAR10C`. Sampled images from the first six clients.

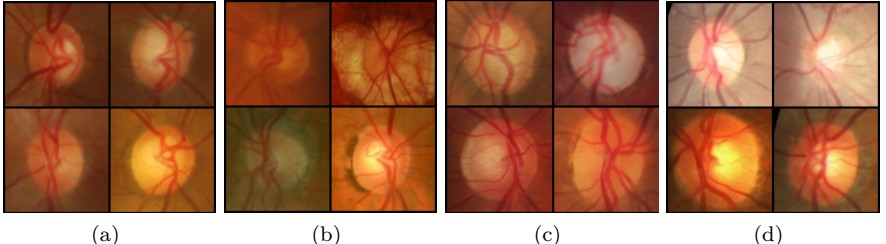

Figure 14: Visualization of the original retina dataset. (a) visualized the Drishti client; (b) visualized the Acrima client; (c) visualized the Rim client; (d) visualized the Refuge client.

## C.3  Visualization of the heterogeneity of the datasets

The visualization of the original distribution in histogram for `DIGITS`, `CIFAR10C`, and `RETINA` images can be found in Figure 18, Figure 19, and Figure 20, respectively.

## C.4  Model architecture

The two model architectures (ResNet18 and ConvNet) are detailed in Table 8 and Table 9, respectively.

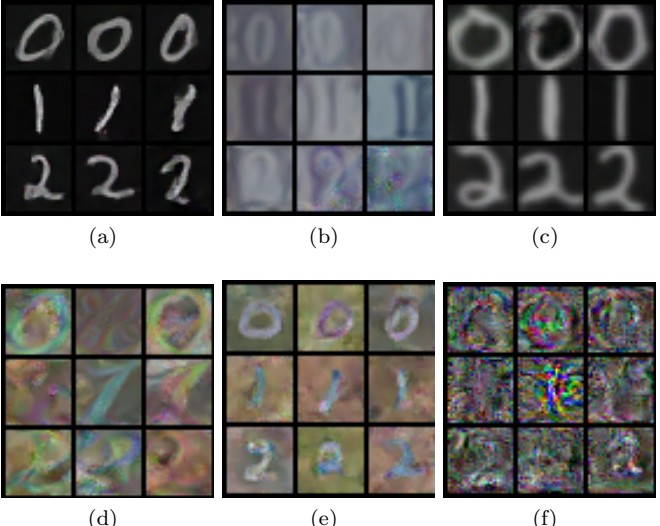

Figure 15: Visualization of the global and local distilled images from the digits dataset. (a) visualized the MNIST client; (b) visualized the SVHN client; (c) visualized the USPS client; (d) visualized the SynthDigits client; (e) visualized the MNIST-M client; (f) visualized the server distilled data.

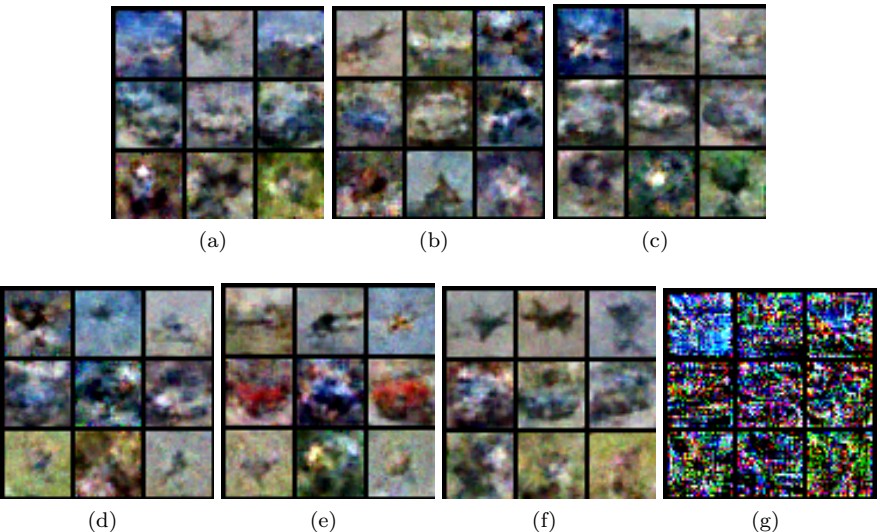

Figure 16: (a)-(f) visualizes the distailled images for the first six clients of CIFAR10C. (g) visualizes the global distilled images.

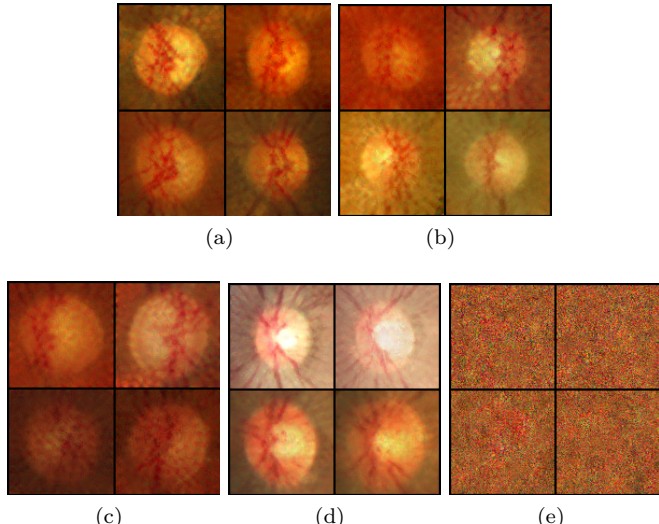

(a)        (b)

(c)        (d)        (e)

Figure 17: Visualization of the global and local distilled images from retina dataset. (a) visualized the Drishti client; (b) visualized the Acrima client; (c) visualized the Rim client; (d) visualized the Refuge client; (e) visualized the server distilled data.

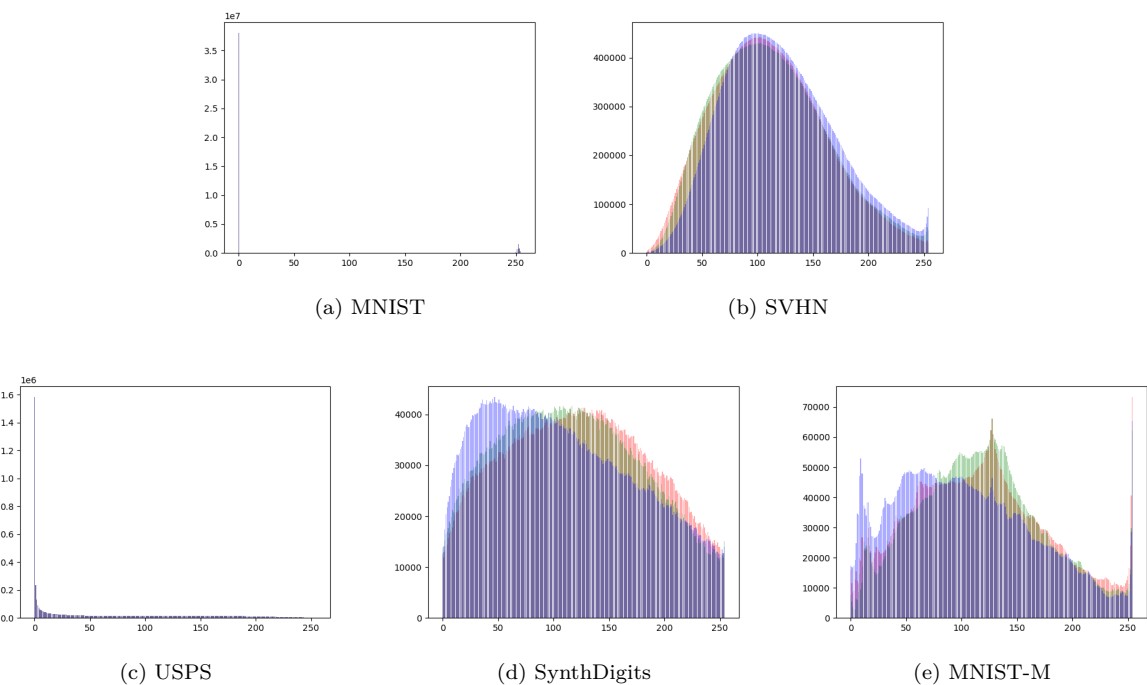

(a) MNIST        (b) SVHN

(c) USPS        (d) SynthDigits        (e) MNIST-M

Figure 18: Histogram for the frequency of each RGB value in original `DIGITS`. The red bar represents the count for R; the green bar represents the frequency of each pixel for G; the blue bar represents the frequency of each pixel for B. One can observe the distributions are very different. Note that figure (a) and figure (c) are both greyscale images with most pixels lying in 0 and 255.

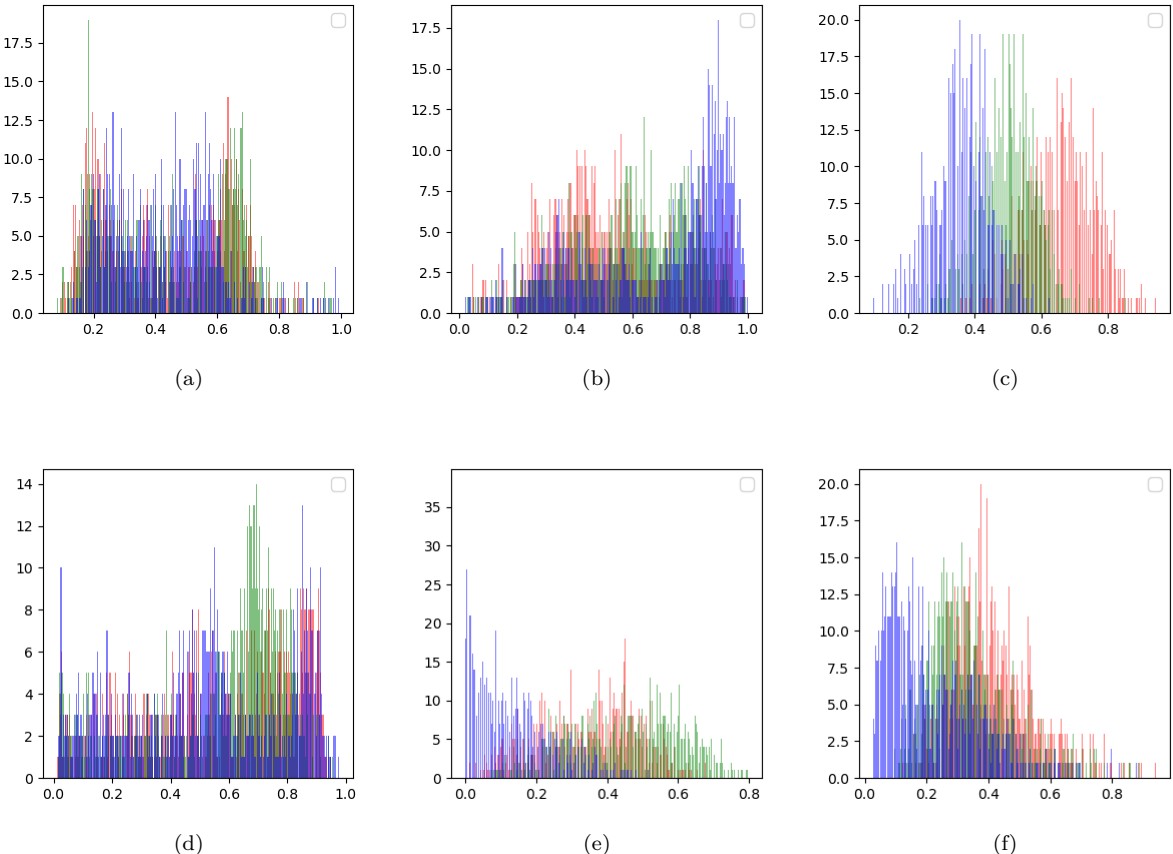

Figure 19: Histogram for the frequency of each RGB value in the first six clients of original `CIFAR10C`. The red bar represents the count for R; the green bar represents the frequency of each pixel for G; the blue bar represents the frequency of each pixel for B.

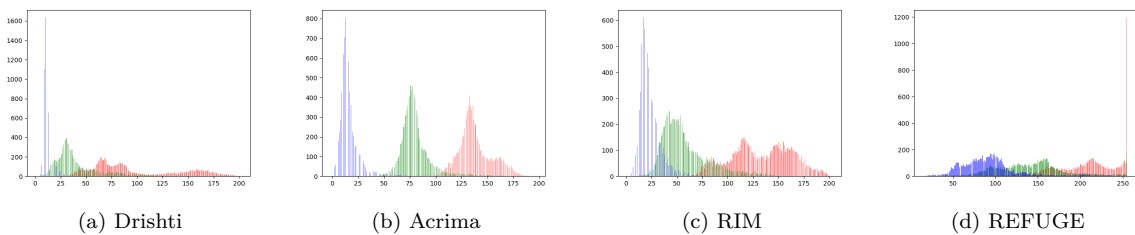

Figure 20: Histogram for the frequency of each RGB value in original `RETINA`. The red bar represents the count for R; the green bar represents the frequency of each pixel for G; the blue bar represents the frequency of each pixel for B.

Table 8: ResNet18 architecture. For the convolutional layer (Conv2D), we list parameters with a sequence of input and output dimensions, kernel size, stride, and padding. For the max pooling layer (MaxPool2D), we list kernel and stride. For a fully connected layer (FC), we list input and output dimensions. For the BatchNormalization layer (BN), we list the channel dimension.

| Layer | Details |
|---|---|
| 1 | Conv2D(3, 64, 7, 2, 3), BN(64), ReLU |
| 2 | Conv2D(64, 64, 3, 1, 1), BN(64), ReLU |
| 3 | Conv2D(64, 64, 3, 1, 1), BN(64) |
| 4 | Conv2D(64, 64, 3, 1, 1), BN(64), ReLU |
| 5 | Conv2D(64, 64, 3, 1, 1), BN(64) |
| 6 | Conv2D(64, 128, 3, 2, 1), BN(128), ReLU |
| 7 | Conv2D(128, 128, 3, 1, 1), BN(64) |
| 8 | Conv2D(64, 128, 1, 2, 0), BN(128) |
| 9 | Conv2D(128, 128, 3, 1, 1), BN(128), ReLU |
| 10 | Conv2D(128, 128, 3, 1, 1), BN(64) |
| 11 | Conv2D(128, 256, 3, 2, 1), BN(128), ReLU |
| 12 | Conv2D(256, 256, 3, 1, 1), BN(64) |
| 13 | Conv2D(128, 256, 1, 2, 0), BN(128) |
| 14 | Conv2D(256, 256, 3, 1, 1), BN(128), ReLU |
| 15 | Conv2D(256, 256, 3, 1, 1), BN(64) |
| 16 | Conv2D(256, 512, 3, 2, 1), BN(512), ReLU |
| 17 | Conv2D(512, 512, 3, 1, 1), BN(512) |
| 18 | Conv2D(256, 512, 1, 2, 0), BN(512) |
| 19 | Conv2D(512, 512, 3, 1, 1), BN(512), ReLU |
| 20 | Conv2D(512, 512, 3, 1, 1), BN(512) |
| 21 | AvgPool2D |
| 22 | FC(512, num_class) |

Table 9: ConvNet architecture. For the convolutional layer (Conv2D), we list parameters with a sequence of input and output dimensions, kernel size, stride, and padding. For the max pooling layer (MaxPool2D), we list kernel and stride. For a fully connected layer (FC), we list the input and output dimensions. For the GroupNormalization layer (GN), we list the channel dimension.

| Layer | Details |
|---|---|
| 1 | Conv2D(3, 128, 3, 1, 1), GN(128), ReLU |
| 2 | AvgPool2d(2,2,0) |
| 3 | Conv2D(128, 118, 3, 1, 1), GN(128), ReLU |
| 4 | AvgPool2d(2,2,0) |
| 5 | Conv2D(128, 128, 3, 1, 1), GN(128), ReLU |
| 6 | AvgPool2d(2,2,0) |
| 7 | FC(1152, num_class) |