# OpenReview forum: "Federated Learning on Virtual Heterogeneous Data with Local-Global Dataset Distillation"
_TMLR — Accepted by TMLR_

### Review · Reviewer_F8EA · 2024-11-01

**Summary Of Contributions:**

This paper presents FedLGD, a method to mitigate data heterogeneity in Federated Learning. FedLGD integrates dataset distillation into FL, using compact synthetic data and aligning local and global data through distribution matching. Experiments show FedLGD outperforms current methods in handling heterogeneous FL settings.

**Audience:**

Yes

**Claims And Evidence:**

Yes

**Requested Changes:**

Following changes are requested, if possible:
1) Change Figure 1 and 4 in a vector form.
2) Extend experiments to larger dataset with more classes (It is also fine, if it is not work in more complicated dataset).
3) Privacy analysis(Optional, I understand it may be difficult).
4) Provide computational cost and communication cost of both local and global training.
5) Consider adding those related works like FedRod, FedGELA, and FedHKD.

**Strengths And Weaknesses:**

Strengths are listed as follows:
1) Strong motivation. The insight that synthetic local data may exacerbate distribution shift is interesting and valuable.
2) Provides a thorough evaluation with both experimental results and theoretical analysis.
3) Well-written and easy to follow.

Weaknesses are listed as follows:
1) Figures 1 and 4 are not vector-based, which could affect clarity and scalability.
2) The experiments are relatively small-scale compared to recent studies in the federated learning literature.
3) The proposed method lacks a privacy analysis about the proposed method, given the authors emphasize the privacy problem in FL.
4) Unlike simpler methods like FedAvg and optimization algorithms such as FedNova, FedProx, and MOON, this approach requires complex computation capabilities on the server.
5) In the related works section, while global aggregation or local training are mentioned, other approaches also address data heterogeneity by combining both strategies. Relevant examples include FedRod, FedGELA, and FedHKD:

FedRod: "On bridging generic and personalized federated learning for image classification"

FedGELA: "Federated learning with bilateral curation for partially class-disjoint data"

FedHKD: "The best of both worlds: Accurate global and personalized models through federated learning with data-free hyper-knowledge distillation"

---

> ### Author Response · Authors · 2024-11-30
> **Response to Weakness 1 - Figure 1 and 4**
>
> We sincerely thank the reviewer for the helpful reminder. We have carefully increased the DPI of the raw figures and saved the processed figures in higher resolution. The updated figures will be included in the revised submission.

---

> ### Author Response · Authors · 2024-11-30
> **Response to Weakness 2 - Justification of the scale of experiments**
>
> We thank the reviewer for the valuable feedback regarding the scale of our experiments. In designing our study, we aimed to address three representative federated learning scenarios: large domain shift (DIGITS), large scale/number of clients (CIFAR10C), and real-world applications with medical images (RETINA).
>
> To address the reviewer’s concern about scale, we have incorporated a larger dataset, CIFAR100C, shown in the table below. Specifically, we generated a federated setting with CIFAR100 by applying 10 distinct corruptions, each representing a client in the federated learning framework. This setup involves 10 federated clients with diverse data distributions. Due to the limited data available per class, we set IPC=10 for local data distillation. The federated learning model was trained over 500 communication rounds, and the resulting average accuracies are as follows:
>
> | FL methods | FedAvg | FedProx | FedNova | Scaffold | MOON | FedProto | VHL    | FedLGD |
> |--------------|--------|----------|--------|--------|--------|----------|--------|--------|
> | Accuracy  | 20.62 | 20.82 | 20.88 | 20.80 | 20.62 | 5.59 | 19.56 | **24.10** |
>
> We believe that incorporating CIFAR100C strengthens the scalability of our experimental evaluation and provides a more comprehensive analysis in line with the federated learning literature. We appreciate the reviewer’s insightful suggestion and welcome any additional feedback to further improve our work.

---

> ### Author Response · Authors · 2024-11-30
> **Response to Weakness 3 - Privacy analysis**
>
> We thank the reviewer for highlighting this critical concern. To address privacy considerations, we provide the following clarifications:
>
> 1. **Empirical Privacy Analysis**: In Section 5.5, we present an empirical evaluation of FedLGD’s privacy defenses, specifically against Membership Inference Attacks (MIAs) and Gradient Inversion Attacks (GIAs). These experiments demonstrate that the proposed method significantly reduces the risk of private information leakage compared to baseline approaches.
>
> 2. **Theoretical Privacy Analysis**: The theoretical privacy benefits of FedLGD are derived from the privacy guarantees of the distilled data, as detailed in references [1] and [2]. These works establish the foundational principles that underpin the privacy-preserving nature of our approach, particularly in scenarios where data reconstruction attacks are a concern.
>
> We hope this addresses the reviewer’s concern, and we welcome additional feedback or suggestions to further enhance the privacy analysis of our method.
>
> [1] Xiong Y, Wang R, Cheng M, Yu F, Hsieh CJ. Feddm: Iterative distribution matching for communication-efficient federated learning. InProceedings of the IEEE/CVF Conference on Computer Vision and Pattern Recognition 2023 (pp. 16323-16332).
>
> [2] Dong T, Zhao B, Lyu L. Privacy for free: How does dataset condensation help privacy?. InInternational Conference on Machine Learning 2022 Jun 28 (pp. 5378-5396). PMLR.

---

> ### Author Response · Authors · 2024-11-30
> **Response to Weakness 4 - Additional computation cost (and RC4)**
>
> We appreciate the reviewer’s thoughtful question. To address concerns regarding computational and communication costs, we clarify the following:
>
> 1. **Global and Local Computational Costs**: Detailed computational costs for both global and local training are presented in Figure 5 of the main text. These include a breakdown of operations required during training iterations, highlighting the added complexity on the server compared to simpler methods like FedAvg.
>
> 2. **Client Communication Costs**: Figure 8 in the appendix provides an analysis of communication overhead. Specifically, it quantifies the cost associated with exchanging updated model parameters or the global virtual data between the server and clients.
>
> While our approach does require higher server-side computation relative to methods like FedAvg, FedNova, and MOON, we argue that this additional complexity enables better convergence properties and robustness to client heterogeneity. These advantages are particularly evident in scenarios with significant non-IID data distributions, as demonstrated in our experiments (Table 1, 2, and 3).
>
> We hope this detailed clarification addresses the reviewer’s concerns. If there are specific aspects of the cost comparison you'd like further elaboration on, we are happy to provide additional details.

---

> ### Author Response · Authors · 2024-11-30
> **Response to Weakness 5 - Additional Related works**
>
> We thank the reviewer for suggesting these relevant works, which we will include in the related work section of our revision. Specifically, we will incorporate them as follows:
>
> “**FedRod** seeks to bridge personalized and generic FL by training separate global and local projection layers. Similarly, **FedGELA** also aims to bridge personalized and generic FL, employing simplex equiangular tight frame (ETF) to address class-imbalance data…… Furthermore, there are strategies to refine the server model or client models using knowledge distillation such as FedDF, FedGen, FedFTG, FedICT, FedGKT, FedDKC, and **FedHKD**. However, we consider knowledge distillation and data distillation two orthogonal directions to solve data heterogeneity issues….”

---

### Review · Reviewer_Q8sp · 2024-11-13

**Summary Of Contributions:**

The paper considers using a distilled virtual dataset, that is smaller than the original client dataset, for federated learning (FL):

- They observe on the USPS and SynthDigits dataset that the wasserstein distance increases between client distributions when distilled, and state that this leads to divergent behaviour in FL when simply replacing client datasets with their distilled version.
- They propose the following method to mitigate the problem, which switches between two modes:
    1. Updating the virtual data: i) the client updates their virtual data using the global feature extractor from the global model ii) the server updates it's global virtual data by using Federated Gradient Matching, where client gradients are used on the server to recover the local dataset
    2. Updating the model: i) clients computes gradients on a cross entropy loss and a contrastive loss (across classes) – both computed on the join global/local virtual dataset ii) the server aggregates gradients and updates the model.
- The communication involved in the algorithms is:
    - clients sends gradients
    - server sends the current model (which includes the feature extractor) and the global virtual dataset

**Audience:**

Yes

**Claims And Evidence:**

No

**Requested Changes:**

As stated under weaknesses in more details, I would suggest the following:

- Include more evidence of data heterogeneity and issue for FL (or downtone the claim)
- Motive the innovations in the algorithm more precisely
- Present the Algorithm more clearly

**Strengths And Weaknesses:**

**Strengths**:

- The paper is including a large number of experiments and comparing with many baselines (7).
- The paper provides a good overview of the literature and cites generously

**Weaknesses**:

One of the main claims is that standard FL with virtual datasets does not work and that this is because of increased data heterogeneity between clients. From Figure 1 and the introduction it appears that this claim is based on experiments on two datasets, one distillation technique and one FL algorithm. To include such a strong claim I think it is necessary to investigate the claim more thoroughly. Specifically:

- The claims "distilled virtual data increases heteogenity" / "the first to reveal that using distilled local virtual data can" seems to only be based on Figure 1. Considering that this is the main motivation for the paper, I suggest providing more evidence for the statement.
    - Does the statement hold up under change of the distillation technique, data, federated learning technique, etc?
    - The evidence of heteogenity is based on tSNE, Wasserstein Distance (37.6%) and Maximum Mean Discrepancy (43.3%). The tSNE I would not trust by itself and WD and MMD are not too informative on high dimensional spaces. I would suggest computing FID instead (i.e. WD on a low dimensional feature space).
- What is the virtual data distillation technique used for Figure 1? The results heavily depends on the process.
- Is the poor performance of FL algorithms on virtual data a question of data heterogeneity or is the reduce performance just a consequence of virtual datasets being of poorer quality, since they are based on a (small) client dataset?
- What prevents us from simply sharing the (smaller) virtual dataset and perform model updates on the server using the joint virtual dataset? This would not have any problems regarding heterogeneity. If this approach also fails it would suggests that data heterogeneity is at least not the only problem.

How the proposed method solves the above problem of virtual data is unclear:

- It is not clear how the contrastive loss in (2) helps with data heterogeneity, since it treats the global/local dataset jointly. The contrastive loss acts across labels, and it is unclear how this relates to resolving the data heterogeneity issue. Is it because the type of heterogeneity considered is label shift, that the method helps?
- Why is the local virtual dataset not directly shared with the server instead of reconstructing it through Federated Gradient Matching?

The presentation of the algorithm could be improved:

- The algorithm is described across section 3.2-3.3 across paragraphs making it hard to follow and. Algorithm 1 in the appendix was critical for me to understand the structure of the proposed method. I suggest pulling it into the main text.
- It is not clear from section 3.2 what is meant by "selected iterations" and why only "early training epochs" is considered. It would be good to reference a precise algorithm at this point.
- Maybe explicitly state that (2) will be optimized over $\theta$.
- explicitly parameterize $\psi$ by  $\theta$ in (2) so it is clear what is being optimizing over.
- "only selecting a few rounds in the early stage of FL is sufficient to synthesize useful global virtual data". Does this mean that the client update of $\tilde D_t^c$ also only happens in the early epochs, since $\tau$ governing the update frequency in Algorithm 1 (appendix) is shared between server and client?

Additional comments:

- Figure 4 tSNE preserved local structures well (clusters) but cannot be used reliably to compare distances between clusters. So it is not too convincing for arguing about a feature shift.
- "In such a way, global virtual data can be served for calibration and groups the features of same classes together". It is not entirely clear to me what this means. Both global and local virtual data plays the same role in the contrastive loss, so why is this comment only considering the global virtual data?

Typos:

- l. 1 morden -> modern
- Section 3.3.2 cliam -> claim

---

> ### Author Response · Authors · 2024-11-30
> **Response to Weakness 1 - Justification for our motivation and the metric we use**
>
> We appreciate the reviewer’s detailed feedback and the suggestions provided to enhance our claims about the impact of distilled virtual data on heterogeneity. Below, we address the specific concerns raised:
>
> 1. **Evidence for the Main Claims**:
>
> The claim that "distilled virtual data *can* increase heterogeneity" is indeed a key motivation of our paper. While Figure 1 provides an initial illustration, our revised analysis will include additional evidence to substantiate this claim:
> - We focus on the **Distribution Matching** approach [2] for distillation and will clarify this explicitly in the manuscript.
>
> - To test the robustness of our findings across different scenarios, we will compute and report the **Maximum Mean Discrepancy (MMD)** score among the DIGITS datasets and the clients (Gaussian noise and fog) from the CIFAR10C as in the attached table. This dataset is characterized by a significant distribution shift, providing a challenging test case for evaluating heterogeneity under various conditions.
>
> By doing so, we aim to demonstrate that the observed exacerbation of heterogeneity persists across a broader range of settings.
>
> 2. **Comparison of Metrics**:
>
> Regarding the choice of evaluation metrics, we selected MMD over Fréchet Inception Distance (FID) as MMD is an **unbiased estimator**, while FID depends on the model under evaluation and may introduce biases [1]. While we understand the reviewer’s suggestion about FID, we believe MMD offers a more principled choice for our context.
>
> 3. **Validation Across Different FL Techniques**:
>
> We acknowledge the importance of validating whether our observations generalize to other federated learning (FL) setups. However, our experiments are conducted before initializing the FL process, making them agnostic to specific FL techniques.
>
> 4. **Clarification on Distillation Techniques in Figure 1**:
>
> The virtual data in Figure 1 is generated using the Distribution Matching-based distillation technique, as noted earlier. We will clearly annotate this in the revised figure caption and corresponding discussion.
>
> We hope these clarifications and planned revisions address the reviewer’s concerns. Thank you for your valuable suggestions, which have helped us strengthen the rigor and transparency of our work.
>
> | FL methods | Original images | Distilled images |
> |--------------|--------|----------|
> | Averaged MMD score on DIGITS  | 0.5794 | 0.8263 |
>
> | FL methods | Original images | Distilled images |
> |--------------|--------|----------|
> | MMD score on sampled CIFAR10C  | 0.4657 | 0.5996 |
>
> [1] Chong MJ, Forsyth D. Effectively unbiased fid and inception score and where to find them. InProceedings of the IEEE/CVF conference on computer vision and pattern recognition 2020 (pp. 6070-6079).
>
> [2] Zhao B, Bilen H. Dataset condensation with distribution matching. InProceedings of the IEEE/CVF Winter Conference on Applications of Computer Vision 2023 (pp. 6514-6523).

---

> ### Author Response · Authors · 2024-11-30
> **Response to Weakness 1 - Explanation on the model performance**
>
> Thank you for highlighting this important point. The performance drop when training with distilled data is indeed a well-documented phenomenon [1,2,3], primarily due to the limited size and inherent approximation of virtual datasets, even in centralized settings. In our paper, we investigate how this challenge is exacerbated in federated learning (FL) due to amplified data heterogeneity across clients.
>
> To address your query about the root cause— the intrinsic quality of virtual datasets versus data heterogeneity—our findings suggest that both factors contribute, but the latter becomes particularly pronounced in FL scenarios. Specifically:
>
> - **Dataset Quality**: The compact nature of virtual datasets does result in reduced representational richness compared to the original datasets. This limitation is consistent across centralized and FL settings.
>
> - **Data Heterogeneity in FL**: The heterogeneity among client datasets further compounds the issue, leading to significant variations in how well the global model adapts to diverse local distributions.
>
> Our proposed approach, FedLGD, mitigates this amplified performance drop by leveraging a supervised contrastive loss with global virtual data and gradually improving the local data quality. This technique effectively aligns representations across heterogeneous data distributions, reducing the adverse impact of both factors. The performance improvements observed in our experiments on DIGITS, RETINA, and CIFAR10C (Tables 1, 2, and 3, respectively) provide evidence of its effectiveness.
>
> We hope this explanation clarifies how the interplay between data quality and heterogeneity influences performance in FL and how our method addresses these challenges.
>
> [1] Zhao B, Mopuri KR, Bilen H. Dataset condensation with gradient matching. arXiv preprint arXiv:2006.05929. 2020 Jun 10.
>
> [2] Zhao B, Bilen H. Dataset condensation with distribution matching. InProceedings of the IEEE/CVF Winter Conference on Applications of Computer Vision 2023 (pp. 6514-6523).
>
> [3] Cazenavette G, Wang T, Torralba A, Efros AA, Zhu JY. Dataset distillation by matching training trajectories. InProceedings of the IEEE/CVF Conference on Computer Vision and Pattern Recognition 2022 (pp. 4750-4759).

---

> ### Author Response · Authors · 2024-11-30
> **Response to Weakness 1 - Justification of *not* sharing local virtual data**
>
> Thank you for raising this insightful suggestion. While sharing the distilled virtual datasets for server-side model updates could indeed address heterogeneity concerns, there are two significant challenges with this approach that we aim to avoid:
> 1. **Increased Communication Overhead**:
>
> Sharing virtual datasets, particularly for image data with large dimensions (e.g., $96 \times 96$), can significantly increase communication costs compared to transmitting model gradients. As shown in Figure 8 of our paper, the communication overhead for virtual datasets can exceed that of neural network models, which is a key concern in federated learning (FL) systems that are often constrained by bandwidth.
>
> 2. **Privacy Concerns**:
>
> Sharing virtual datasets introduces a potential for privacy leakage that is not adequately addressed by current privacy analyses. Existing studies typically focus on the privacy of models trained on distilled virtual datasets rather than the datasets themselves [1].
> In extreme cases, such as distilling 50 real data points into 50 virtual ones, achieving a sufficient privacy guarantee (e.g., differential privacy) for the distilled data would require a prohibitively high privacy budget. This challenge becomes particularly critical in scenarios with sensitive client data.
>
> Given these considerations, we propose leveraging the pre-existing gradient-sharing mechanism of classic FL instead of directly sharing virtual datasets. This approach minimizes additional communication overhead and avoids introducing new privacy risks while still addressing heterogeneity through techniques such as supervised contrastive loss (as discussed in the paper).
> We hope this explanation clarifies our reasoning and demonstrates the trade-offs involved in the alternative approach you suggested.
>
> [1] Dong T, Zhao B, Lyu L. Privacy for free: How does dataset condensation help privacy?. InInternational Conference on Machine Learning 2022 Jun 28 (pp. 5378-5396). PMLR.

---

> ### Author Response · Authors · 2024-11-30
> **Response to Weakness 2 - How the supervised contrastive loss works**
>
> Sorry for the confusion. We believe the confusion stems from the misunderstanding of the heterogeneity types in our work. We also wanted to clarify, that the supervised contrastive loss does NOT act across labels, but acts per class, as in the explanation of our Eq. 4, “$B^{y_j}_{\backslash j}$ is a subset of $B_{\backslash j}$ only with samples belonging to class $y_{j}$”.
>
> 1. **Type of Heterogeneity Considered**:
> - The primary type of heterogeneity addressed in our work is **domain shift** rather than **label shift**, represented by variations in $P(X|y)$ across clients, where $X$ denotes input data and $y$ the corresponding labels. This differs from label shift ($P(y)$) and is more relevant to our use case, as the variations in data distributions across clients significantly impact feature representation and model performance.
> - To handle the heterogeneity issue, our goal is to enforce similarity in feature representations of the same class, thus we leverage the supervised contrastive loss to mitigate the domain shift within each class and ensure that models trained on different local datasets align effectively with the global representation.
>
> 2. **Role of Supervised Contrastive Loss**:
> - Recalling that we want to regularize the alignment in the feature space for the domain shift FL problem, the supervised contrastive loss is designed to align feature representations by decreasing feature distances for data points. Meanwhile, we need the data feature to be distinguished across different classes, thus the supervised contrastive loss is performed to decrease the distance from the same class while increasing the distance for different classes. This alignment helps reduce discrepancies in feature spaces across clients, even in the presence of heterogeneity.
> - We can detach the global virtual data during back propagation, which facilitates the generalization of FL.  In our implementation, we chose not to detach the global virtual data from the loss function. This decision allows the global virtual data to directly influence local client models, improving both generic FL performance (via shared global representations) and personalized FL performance (by preserving some local-specific characteristics).
>
> To clarify these points, we will provide a more detailed explanation in the revised manuscript, including a formal definition of the domain shift addressed, the specific role of global virtual data, and how the supervised contrastive loss aids in mitigating these shifts.
>
> [1] Khosla P, Teterwak P, Wang C, Sarna A, Tian Y, Isola P, Maschinot A, Liu C, Krishnan D. Supervised contrastive learning. Advances in neural information processing systems. 2020;33:18661-73.

---

> ### Author Response · Authors · 2024-11-30
> **Response to Weakness 3 - The presentation of the manuscript**
>
> - We thank the reviewer for the concrete suggestion. We will move the algorithm box to the main text for a more straightforward presentation.
>
> - We sample “selected iterations” from “early training epochs” because they contain larger gradient information, and the larger gradient information can facilitate better gradient-based image synthesis [1]. We appreciate the reviewer’s insightful suggestion and will add the reference and motivation in our revision.
>
> - We will explicitly state $\psi$ with $\theta$ for more clear presentation.
>
> - Yes, we update both $\tilde{D}^c_t$ and $\tilde{D}^g_t$ only on the selected iterations.
>
> [1] Huang Y, Gupta S, Song Z, Li K, Arora S. Evaluating gradient inversion attacks and defenses in federated learning. Advances in neural information processing systems. 2021 Dec 6;34:7232-41.

---

### Review · Reviewer_SoTu · 2024-11-17

**Summary Of Contributions:**

This paper addresses the issue of data heterogeneity across clients in federated learning via local-global data distillation.

**Audience:**

Yes

**Claims And Evidence:**

No

**Requested Changes:**

- cliam -> claim
- the consistency of the symbols
- Big bracket for equation 5 and 6
- i.e.,, -> i.e.,
- What does `different images per class` mean?

**Strengths And Weaknesses:**

Strengths:
- The idea of local-global dataset distillation is interesting.
- The authors conducted extensive experiments to demonstrate the effectiveness of their method

Weakness:
- The symbols are not consistent throughout the paper, i.e., $\tilde{D}_i$, $\tilde{D}^c$, $\tilde{D}_g$
- The definition of the small virtual dataset is unclear. More specifically, why does the equation $|\tilde{D}_i| \ll |D_i|$ need to hold?
- The authors claim that `We cliam a ‘good’ global virtual data should be representative of the global data distributions`. If I understand correctly, in most existing FL literature, the global data is typically considered the test dataset. Could the authors elaborate on a few scenarios or works where the global data exists but is not the test dataset?
- In the experimental results section, it is surprising to see that FedAvg outperforms other more advanced methods, which are specifically designed to address data heterogeneity, in most experimental setups. The authors are encouraged to explain the reasons behind this.
- According to Figure 2, in addition to the local model and the global model being shared between clients and the server, the global virtual data $\tilde{D}_{new}^g$ is also shared between the clients and the server. However, in the Privacy section, the authors mention that `We note that FedLGD uses pre-existing information, i.e.,, shared averaged gradients and global model, to distill virtual data, so there is no extra privacy leakage` . The authors are encouraged to explain the discrepancy between these two parts.
- The authors are also encouraged to explain why FedLGD has better privacy protection than FedAvg.
- The technical contribution is marginal.
- The authors mention that `The global virtual data then serves as anchors to alleviate domain shifts among clients.` Could the authors indicate which experimental results support this statement? Additionally, the authors need to define "domain shift" as used in this paper prior to making this statement.

---

> ### Author Response · Authors · 2024-11-30
> **Response to Weakness 1 - Inconsistent symbols**
>
> Thank you for pointing this out. We sincerely apologize for the inconsistency in the use of symbols throughout the paper. Specifically:
>
> - $D_i$: Refers to the local dataset for client i.
> - $D^c$: Refers to the collection of local datasets. Specifically, $D^{c_i}$ refers to the local dataset for client i.
> - $\tilde{D}^c$: Refers to the collection of local virtual datasets. Specifically, $\tilde{D}^{c_i}$ refers to the local dataset for client i.
> - $\tilde{D}^g$​: Refers to the global dataset and the distilled global virtual dataset, respectively.
>
> We notice that we have inconsistent use of $D_i$ and $D^c$, and we have carefully reviewed and standardized the notation to ensure consistency across all sections. We have updated the manuscript to use these symbols consistently and clearly define them upon first use to avoid any confusion. Thank you for bringing this to our attention, and we appreciate your feedback in helping us improve the paper.

---

> ### Author Response · Authors · 2024-11-30
> **Response to Weakness 2 - Clarification of the definition of the small virtual dataset**
>
> Thank you for highlighting this point.
> To address your question, we would like to begin by highlighting one of the motivations of our proposed method, as we stated in our introduction – improving the efficiency of training in FL compared with using original data $D_i$. Therefore, we aim to obtain a highly compact representation $\tilde{D}_i$ of the original dataset ($D_i$), which has **a significantly smaller number of data samples while retaining its essential information**. Such datasets are achieved via data distillation, and we refer to them as small virtual datasets. Concretely, the two desired characteristics of distilled data are:
>
> - **Efficiency**: A smaller distilled dataset significantly reduces the computational and communication overhead during training, which is especially important in resource-constrained environments like federated learning.
>
> - **Representation Quality**: The distillation process aims to encode rich and representative information from $D_i$​ into $\tilde{D}_i$​. By design, the distilled dataset is optimized to serve as an effective surrogate for training, capturing the key characteristics of $D_i$ despite its reduced size.
>
> Thus, it is intuitive to see that if |$\tilde{D}_i$|≪|$D_i$| does not hold, we can not achieve the efficiency goal. We hope this clarification provides a better understanding of the rationale behind this design choice. Thank you again for your valuable feedback!

---

> ### Author Response · Authors · 2024-11-30
> **Response to Weakness 3 - Explanation of the claim of global virtual data**
>
> Thank you for raising this point. This provides the opportunity to further highlight the advantage of our FedLGD’s flexibility on global data assumptions.
>
> As you mentioned, in most existing FL setups, global data is often synonymous with the test dataset. We claim that the testing data is not always feasible or suitable as the global data, for example when the size of testing data is extremely small or it comes from another domain. In addition to this global data assumption, some recent FL methods, such as FedDF [1] and FCCL [2], leverage externally collected global data to facilitate training. These approaches assume the availability of in-domain global data during training, which may not always be feasible in real-world scenarios.
>
> In contrast, FedLGD eliminates the need for this assumption by proposing to distill global virtual data directly from shared gradients. This approach allows us to approximate the global data distribution without requiring explicit access to global data. This makes FedLGD more applicable in practical settings where collecting in-domain global data is challenging or impossible.
>
> We hope this clarification highlights the distinctiveness of FedLGD and its ability to overcome limitations in scenarios requiring global data for training. Thank you for the opportunity to elaborate further.
>
> [1] Lin T, Kong L, Stich SU, Jaggi M. Ensemble distillation for robust model fusion in federated learning. Advances in neural information processing systems. 2020;33:2351-63.
>
> [2] Huang W, Ye M, Du B. Learn from others and be yourself in heterogeneous federated learning. InProceedings of the IEEE/CVF Conference on Computer Vision and Pattern Recognition 2022 (pp. 10143-10153).

---

> ### Author Response · Authors · 2024-11-30
> **Response to Weakness 4 - Justification for FedAvg results**
>
> Thank you for your observation. First, it is no surprise that FedAVG outperforms more complex methods. It is worth noting that this phenomenon is not unprecedented in the literature. For example, FedAvg has demonstrated competitive performance in experiments reported in [1], which shares similar problem settings to ours, even when compared to methods designed to address data heterogeneity.
>
> Coming back to our results, a possible explanation for FedAvg outperforming more advanced methods in certain setups could be the limited amount of local data available. Advanced methods often rely on more complex mechanisms, which may not offer a significant advantage in such scenarios. For instance, FedProx’s regularization term primarily mitigates divergence when the local model deviates substantially from the global model, a situation that may not always arise in our experimental conditions due to the compact size of local virtual data.
>
> [1] Tang Z, Zhang Y, Shi S, He X, Han B, Chu X. Virtual homogeneity learning: Defending against data heterogeneity in federated learning. InInternational Conference on Machine Learning 2022 Jun 28 (pp. 21111-21132). PMLR.

---

> ### Author Response · Authors · 2024-11-30
> **Response to Weakness 5 - Justification for the information shared by $\tilde{D}^g$**
>
> Thank you for your question. Sorry for the confusion, we believe there may have been a misunderstanding, and we would like to clarify the process regarding $\tilde{D}^g$. The distilled dataset $\tilde{D}^g$ is derived exclusively using information from the shared global model. Therefore, sharing $\tilde{D}^g$ does not introduce any additional privacy risks beyond those already inherent in sharing the global model itself. This is because anyone with access to the global model can independently distill the same virtual dataset $\tilde{D}^g$ using the same methodology.

---

> ### Author Response · Authors · 2024-11-30
> **Response to Weakness 6 - Privacy protection of FedLGD**
>
> Thank you for the question and we are pleased to clarify the contribution of FedLGD.
>
> We begin by introducing the potential privacy attacks in FL: membership inference attack and gradient inversion attack are two essential concerns that data is inferred from the shared information (e.g., gradients and logits) **directly** derived from clients’ data.
>
> The ability for better privacy protection lies in the **virtual data** FedLGD used for training. In contrast to vanilla FedAvg, which is trained on real data and shares gradient w.r.t. real data, FedLGD relies on the locally distilled virtual data to perform training on the clients’ sides instead of the local raw data. By doing this, loca data information is protected by the data distillation process. It has been shown in the literature that training with distilled virtual data can defend from Membership Inference Attacks and Gradient-based Inversion Attacks [r1]. In our manuscript, we empirically show the defense efficacy of FedLGD in Figure 6 and 7.
>
> [r1] Dong T, Zhao B, Lyu L. Privacy for free: How does dataset condensation help privacy?. InInternational Conference on Machine Learning 2022 Jun 28 (pp. 5378-5396). PMLR.

---

> ### Author Response · Authors · 2024-11-30
> **Response to Weakness 7 - Justification of the technical contribution**
>
> We respectfully disagree with the assessment that the technical contribution is marginal and would like to justify the novelty and difficulty of our problem setting.
>
> 1. **Novel and Unexplored FL setting**:
>
> In this work, we focus on Federated Virtual Learning (FVL), a new FL frame and promising approach to applying data distillation-based synthetic data to FL pipeline to handle label shift, asynchronization, and privacy problems in FL. We propose FedLGD, which incorporates iterative local and global data distillation to achieve good performance with limited amounts of distilled virtual data.  We proposed Iterative Distribution Matching to inpaint the global information to local virtual data using the up-to-date global model. Through local virtual data distillation, *class-balanced* synthetic data are generated to facilitate FL training.
>
> 2. **Novel, effective, and efficient pipeline**:
>
> As the very first work leveraging distillation-based synthetic data in FL training, we develop a novel pipeline that tackles the challenges around effectiveness and efficiency. Particularly, we carefully design different distillation methods and loss functions in our proposed pipeline:
> - *Overcoming Challenges of Distribution Matching in FL*: We observe that existing dataset distillation techniques, such as Distribution Matching, may unintentionally exacerbate data heterogeneity among clients. FedLGD introduces a novel local-global data distillation framework to tackle this issue by iteratively updating and aligning local and global virtual data. This approach effectively regularizes heterogeneity while maintaining compact and efficient representations.
> - *Addressing Efficiency in Bi-level Optimization with Efficient Global Virtual Data Distillation*: We proposed Federated Gradient Matching to efficiently incorporate the bi-level optimization problem of data distillation using gradient matching into the classical FL pipeline. The virtual global data could be used to regularize feature heterogeneity among clients.
>
> 3. **Theoretical and Empirical evidence**: Through our theoretical analysis and the comprehensive experiments on benchmark and real-world datasets, we showed that FedLGD outperformed existing state-of-the-art FL algorithms.
>
> We believe these contributions collectively address significant challenges in federated learning and represent a meaningful step forward. Thank you for the opportunity to clarify.

---

> ### Author Response · Authors · 2024-11-30
> **Response to Weakness 8 - Clarification of the domain shift and how FedLGD alleviates it**
>
> Thank you for your thoughtful comment. FedLGD alleviates performance drops caused by domain shifts by utilizing a supervised contrastive loss with global virtual data, which acts as anchors to align distributions among clients. This is evidenced by the performance improvements in our experiments on the DIGITS, RETINA, and CIFAR10C datasets (Tables 1, 2, and 3, respectively), where the global model trained with FedLGD achieves the highest average test accuracy.
>
> We appreciate your suggestion to define "domain shift" more clearly. In our revision, we will explicitly define domain shift as variations in $P(X|y)$ across clients, where $X$ represents the input data and $y$ the corresponding labels. This definition will provide a clearer context for our statement and the role of global virtual data in mitigating these shifts.
>
> Thank you for bringing this to our attention, as it helps enhance the clarity and rigor of our paper.

---

> ### Author Response · Authors · 2024-11-30
> **Response to Requested changes 5 - Meaning of different images per class**
>
> Images Per Class (IPC) is a standard term in image-based dataset distillation, particularly for classification tasks. It represents the number of virtual images distilled for each class. The total size of the distilled dataset is calculated as the product of the number of classes and the IPC (num_class×IPC\text{num\_class} \times \text{IPC}num_class×IPC). IPC thus provides a practical way to define and control the size of the distilled dataset and serves as an essential hyperparameter in the optimization process for distillation.
>
> In Table 1 of our paper, we experiment with different IPC values (e.g., 10 and 50) to adjust the sizes of the distilled datasets in the DIGITS experiment. This variation allows us to analyze the impact of IPC on model performance.

---

> > ### Comment · Reviewer_SoTu · 2024-12-16
> >
> > I thank the authors for their response. Many issues have been addressed, but I still have some remaining questions:
> >
> > - Answer 1: Does this mean that $D_i$​ is essentially the same as $D^{c_i}$​? If they are the same, I suggest removing one of the notations to avoid confusion. Additionally, $\tilde{D}^{c_i}$ should represent the local virtual data on client $i$, rather than the local data on client $i$, correct? Lastly, the client index seems to be missing in Algorithm 1. Is this intentional?
> >
> > - Answer 3: I understand and agree with the authors that collecting an in-domain global test dataset can sometimes be challenging. However, I still have some questions. You stated, _"We claim a ‘good’ global virtual data should be representative of the global data distributions. Therefore, we propose to leverage local clients’ averaged gradients to distill global virtual data."_ This implies that local averaged gradients can effectively represent the global data distribution. Does this assumption hold universally? Furthermore, since the generated virtual data is heavily influenced by the training data through gradients, isn’t there a risk that the model could overfit to this generated global virtual data? In such a case, can the global virtual data still be considered representative of the real-world global data distribution?

---

> > > ### Author Response · Authors · 2024-12-20
> > > **Response to the comment on answer 1**
> > >
> > > We appreciate the reviewer’s insightful observations and follow-up questions. We recognize the importance of maintaining clarity in notation and thank you for highlighting these points.
> > >
> > > 1. **On $D_i$ and $D^{c_i}$​:** Yes, $D_i$ and $D^{c_i}$​ are indeed the same. To avoid confusion, we have revised the manuscript to consistently use $D^{c_i}$​ throughout.
> > > 2. **On $\tilde{D}^{c_i}$:** You are correct; $\tilde{D}^{c_i}$ represents the local virtual data for client i, not the local data directly available on the client.
> > > 3. **On the client index in Algorithm 1:** The omission of the client index in Algorithm 1 was intentional, aimed at simplifying the presentation. Since the operations are identical across clients, we chose a general form to reduce visual complexity.
> > >
> > > We are grateful for the opportunity to improve the manuscript’s clarity and presentation based on your valuable feedback. Thank you again for your thoughtful comments.

---

> > > ### Author Response · Authors · 2024-12-20
> > > **Response to the comment on answer 3**
> > >
> > > > Clarification on Capturing Global Data Distribution from Averaged Gradients
> > >
> > > We sincerely thank the reviewer for their insightful comments. In FL, the global data distribution is conceptualized as the joint or mixture of the local data distributions [r1]. The averaged gradient, as used in our approach, is a principled proxy for the gradient derived from this global data. Building on our theoretical analysis, we show that under reasonable assumptions  (Lemma 2), data distillation through gradient matching serves as an effective approximation of distribution matching. Moreover, via optimizing our designed loss functions, we can minimize the statistic margins between global virtual data vs local training data, and between local virtual data vs local real data.  This two-fold optimization ensures that the global virtual data distilled from averaged gradients aligns closely with the underlying joint distribution of local datasets. While we acknowledge that theoretical assumptions may not always translate perfectly to practice, our extensive experiments across diverse datasets and model architectures consistently demonstrate the efficacy of this approach. In every setting, our method reduces statistical margins in FL, reinforcing the representativeness and utility of the global virtual data.
> > >
> > > > Clarification on Addressing Overfitting Concerns and Ensuring Representativeness
> > >
> > > We sincerely thank the reviewer for bringing up the important question regarding potential overfitting to the global virtual data. We would like to emphasize that the risk of overfitting in our approach is inherently minimal due to the following carefully designed aspects of our framework:
> > >
> > > 1. **Limited Influence of Global Virtual Data:**  In our training strategy, the contribution of global virtual data is intentionally limited relative to local data. For instance, in experiments on DIGITS and CIFAR10C, the ratio of global to local virtual data is maintained at 1:5. This ensures that training remains predominantly influenced by local data, inherently mitigating risks of overfitting to the global virtual data.
> > >
> > > 2. **Early-Stage Regularization with Non-Overfitted Gradients:** To harmonize the representations of heterogeneous clients and mitigate overfitting, we introduce a regularization mechanism that leverages global virtual data distilled from the gradients of  non-overfitted models captured starting from the early stages of training. At this stage, the model has not yet converged or overfitted to local data, ensuring that the global virtual data retains broad representativeness. By anchoring the training process with these initial gradients, we mitigate the risk of later-stage overfitting and limit the influence of global virtual data derived from potentially overfitted models.
> > >
> > > These mechanisms are carefully designed not only to reduce the risk of overfitting but also to reinforce the representativeness of the global virtual data.
> > >
> > > [r1] ​​​​​​Deng W, Thrampoulidis C, Li X. Unlocking the potential of prompt-tuning in bridging generalized and personalized federated learning. InProceedings of the IEEE/CVF Conference on Computer Vision and Pattern Recognition 2024 (pp. 6087-6097).

---

### Review · Reviewer_ye5S · 2024-11-18

**Summary Of Contributions:**

The authors present a pipeline that uses distilled virtual dataset, which represents a smaller and condensed form of the training data for Federated Learning (FL):
1. Authors use distribution matching (DM) [zhao and Bilen, 2023] to match the model's feature distribution on the original dataset $D^c$ and distilled dataset $\tilde{D}^c$ in an iterative manner.
2. They propose adding a regularization term in the feature space to the total loss function to handle data heterogeneity among clients arising due to data distillation.
3. Gradient matching is used to distill global virtual data to make the global virtual data representative of the global data distribution.

**Audience:**

Yes

**Claims And Evidence:**

No

**Requested Changes:**

1. Fix typos and improve writing/presentation.
2. Either CIFAR-100 or Tiny ImageNet would serve as better test cases to validate the ideas at a larger scale and with more classes. If this is not possible, please either:
    1. Provide convincing arguments for why CIFAR-10-C qualifies as a "large-scale dataset" or,
    2. Revise claims about the method's effectiveness on large-scale datasets.
3. Please, either compare to related distillation works [1,2] or provide arguments why it's not necessary.
4. Please provide more justification (e.g. experimental) to the claim: "we are the first to reveal that using distilled local virtual data can exacerbate the heterogeneity problem".

**Strengths And Weaknesses:**

### Strengths
1. The idea of harmonizing local heterogeneity with global anchors is an interesting one.
2. Decent literature review.

### Weaknesses
Notational issues:
1. In description of eq (1), $x_i$ and $\tilde{x}_j$ are never introduced, including in the notation section of the appendix. The reader is left to assume it's meaning.

Claims and explanations:
1. Authors claim that they are "the first to reveal that using distilled local virtual data can exacerbate the heterogeneity problem". However, only one example is provided if I am not mistaken with USPS and SynthDigits. Does this hold in Cifar10 and other dataset the authors use? I think more is needed to justify this claim.
3. Authors mention that their choice to use global feature extractors instead of randomly initialized models to extract features in turn generates better task specific local virtual data and that this is inspired by empirical neural tangent kernel works. Could you explain how are these related?

Experimental Setup:
1. Relatively simpler tasks and datasets are used (retina with binary classification, cifar10c, and digits). While this is not necessarily a critique, in does limit the claims in the paper, such as "we evaluate the performance of FedLGD on another large benchmark dataset"
2. Although the authors compare with a range of FL methods in their baselines, none of the methods use distillation. Some distillation in FL works [1, 2]. Could the authors either compare to these works or provide arguments on why comparison is not necessary?


Writing and Language:
1. The writing fell short for me. While this might be acceptable in some cases, here it significantly reduces the clarity of the exposition.
2. Too many adjectives are jumbled together to create noun phrases, often multiple times in single sentences, making the paper very hard to read - especially in crucial places. For example: "... address the heterogeneity issues with the aid of *global virtual anchor distillation and regularization* ..." This is a common theme across the paper.
3. Too many typos. E.g.
    - "With compromising privacy in implementation, our global anchors ...". Should it be *without compromising*?
    -  "morden" machine learning. etc.

Additional Comments and Questions:
1. Is the idea of global anchors similar to global prototype distillation [2]? Does it (or not) differ?
2. What does Images Per class (IPC) mean?

**References:**
1. Federated Learning via Decentralized Dataset Distillation in Resource-Constrained Edge Environments [2023].
2. Global prototype distillation for heterogeneous federated learning [2024].

---

> ### Author Response · Authors · 2024-11-30
> **Response to Weakness 1 and RC 1 - notation issues**
>
> Thanks for the question, $x_i$ and $\tilde{x}_j^t$ are the data sampled from $D^c_k$ and  $\tilde{D}^{c,t}_k$, respectively. We have clarified them and added them in our notation section in our revision.

---

> ### Author Response · Authors · 2024-11-30
> **Response to Weakness 2 and RC 4 - Clarification and explanation for our heterogeneity statement**
>
> We appreciate the reviewer’s thoughtful comment and the opportunity to clarify our contributions.
>
> In fact, we consistently observe similar patterns across various cases. To provide more evidence, we report MMD scores among DIGITS datasets and clients (Gaussian noise and fog) from the CIFAR10C below.
> | FL methods | Original images | Distilled images |
> |--------------|--------|----------|
> | Averaged MMD score on DIGITS  | 0.5794 | 0.8263 |
>
> | FL methods | Original images | Distilled images |
> |--------------|--------|----------|
> | MMD score on sampled CIFAR10C  | 0.4657 | 0.5996 |
>
> Also, we would like to clarify that our statement that "using distilled local virtual data *can* exacerbate the heterogeneity problem" aims to reveal the observation that the amplified distribution shift can exist, which no one thought of this phenomenon before, thus we only used one example to deliver the information in the introduction to indicate the existance.
>
> However, We recognize that additional evidence could further strengthen this claim. Following your suggestions, we have provided a more precise description of our dataset distillation methodology, its implications for heterogeneity, and more examples in our revision. Thank you again for your insightful feedback; it has been instrumental in enhancing our work.

---

> ### Author Response · Authors · 2024-11-30
> **Response to Weakness 2 - Explanation of how the feature extractor was inspired by the empirical neural tangent kernel**
>
> We appreciate the reviewer’s insightful question. The connection between our approach and empirical Neural Tangent Kernel (NTK) theory lies in the way we extract features using **trained** models. The original NTK theorem assumes that models are overparameterized, and the feature derived by the random weights can capture the learning dynamic of neural networks. However, in practice, neural networks do not meet the overparameterization assumption. Especially in the FL settings, the models are typically not big for computation and communication efficiency consideration, limiting the direct application of NTK using random model weights.  However, prior empirical NTK work has shown that a trained model, as opposed to a randomly initialized one, can more effectively capture NTK features with small models, improving performance in both prediction and generative tasks [1,2]. Building on this insight, we use the evolving global feature extractor as our empirical NTK kernel, which allows for more accurate representation learning. This facilitates the generation of task-specific local virtual data, leveraging the benefits of NTK-inspired feature extraction without requiring an overparameterized model to meet the realistic settings in FL.
> We hope this explanation clarifies the relationship between our approach and NTK theory. Thank you for the opportunity to elaborate.
>
>
> [1] Arora S, Du SS, Li Z, Salakhutdinov R, Wang R, Yu D. Harnessing the power of infinitely wide deep nets on small-data tasks. arXiv preprint arXiv:1910.01663. 2019 Oct 3.
>
> [2] Yang Y, Adamczewski K, Sutherland DJ, Li X, Park M. Differentially private neural tangent kernels for privacy-preserving data generation. arXiv preprint arXiv:2303.01687. 2023 Mar 3.

---

> ### Author Response · Authors · 2024-11-30
> **Response to Weakness 3 and RC 2 - Highlights on the comprehensive dataset used in the evaluation**
>
> **1. Highlights the complexity of the tasks included in our submission**
>
> We sincerely thank the reviewer for highlighting this important point and for the valuable suggestion. We would like to clarify that our experimental design was carefully crafted to address three widely studied and challenging federated learning scenarios: (1) large domain shifts, demonstrated using the DIGITS dataset; (2) large-scale settings with a significant number of clients and dataset corruptions, illustrated by CIFAR10C, which includes 57 corrupted versions of CIFAR10, resulting in a total dataset size of 60k × 57 images; and (3) real-world medical imaging, represented by the RETINA dataset, which consists of high-resolution (96 × 96) medical images for binary classification.
>
> While these datasets serve as practical benchmarks for their respective scenarios, we acknowledge the reviewer's concern regarding task complexity. In response to this insightful feedback, we have incorporated CIFAR100C into our experiments (shown in the attached table in our next response), which introduces a larger dataset and a more challenging multi-class classification task, further strengthening our claims and extending the robustness of our evaluation.
>
> **2. Additional results on Cifar100-C**
>
> Thank you once again for your constructive comments, which have helped us enhance the scope and rigor of our work.
> CIFAR100C experiment setup: We sampled CIFAR100 with 10 corruptions (where each corrupted CIFAR100 set represents a client in federated learning), resulting in 10 federated learning clients. Given the limited data per class, we set ipc=10 for local data distillation. We trained the federated learning model for 500 rounds, and the resulting average accuracies are as follows:
>
>  | FL methods | FedAvg | FedProx | FedNova | Scaffold | MOON | FedProto | VHL    | FedLGD |
> |--------------|--------|----------|--------|--------|--------|----------|--------|--------|
> | Accuracy  | 20.62 | 20.82 | 20.88 | 20.80 | 20.62 | 5.59 | 19.56 | **24.10** |

---

> ### Author Response · Authors · 2024-11-30
> **Response to Weakness 3 and RC 3 - Discussing the differences of the additional related works using ‘distillation’**
>
> Thanks for the comments. We would like to point out that the suggested literature does not align with our setting or fall into the same categories as our techniques. We begin by restating our settings: 1) no direct or indirect data sharing from local to global; 2) we focus on data distillation methods rather than knowledge (model) distillation, an orthogonal technique in FL. Our problem setting involves training a global model in a classical FL scenario using local virtual data, where FedLGD aims to address data heterogeneity through the proposed local-global data distillation. Importantly, in our experiments, all baselines use distilled data for local training to ensure a fair comparison.
>
> We thank the reviewer for highlighting these two interesting papers and provide the following clarifications regarding their relevance:
>
> **1. Different Settings with [1]: Directly Sharing Locally Distilled Data**
>
> The approach in [1] involves directly sharing locally distilled data across clients, which differs fundamentally from our framework. This approach raises two concerns:
> - Increased Communication Overhead: Sharing large-sized distilled data (e.g., 96 × 96 images) can significantly increase communication costs compared to sharing NN model parameters (refer to Figure 8).
> - Privacy Risks: Current privacy analyses focus on models trained on distilled data, not the data itself, leaving potential privacy leakage risks unaddressed [r1]. Therefore, [1] operates in a different setting and is not directly comparable.
>
> **2. Different Settings with [2]: Dataset Distillation vs. Knowledge Distillation**
>
> The method in [2] does not employ dataset distillation but rather focuses on knowledge distillation, and its global prototype generation strategy aligns more closely with frameworks like FedProto [r2], as noted in our Related Work section. While knowledge distillation addresses heterogeneity through prototype sharing, it represents an orthogonal strategy to our data-distillation-based approach, making a direct comparison less relevant.
>
> We hope this clarifies why these works are outside the scope of our comparative analysis. Thank you for your thoughtful feedback, which has allowed us to better position our contributions.
>
> [r1] Dong T, Zhao B, Lyu L. Privacy for free: How does dataset condensation help privacy?. InInternational Conference on Machine Learning 2022 Jun 28 (pp. 5378-5396). PMLR.
>
> [r2] Tan Y, Long G, Liu L, Zhou T, Lu Q, Jiang J, Zhang C. Fedproto: Federated prototype learning across heterogeneous clients. InProceedings of the AAAI Conference on Artificial Intelligence 2022 Jun 28 (Vol. 36, No. 8, pp. 8432-8440).

---

> ### Author Response · Authors · 2024-11-30
> **Response to Weakness 4 and RC 2 - Writing and language**
>
> Thank you for your thoughtful feedback on the writing. We are sorry for the confusion caused for you.  Since the paper introduces a novel approach utilizing distilled local and global virtual data for training, we initially aimed to provide detailed descriptions for clarity. Recognizing such confusion might be caused by overly complex sentences and stacked adjectives, we have tried our best to rephrase them to improve clarity.   For example, we rephrased the sentence to: “To overcome the limitations, we propose an effective solution to address the heterogeneity issues using global virtual anchor for regularization, supported by our theoretical analysis.” Additionally, we have carefully revisited the manuscript to rephrase sentences with excessive adjectives or lengthy constructs, ensuring a smoother and more accessible reading experience throughout.
>
> We believe that our careful rephrasing of the long sentences has addressed your comments, and we are pleased to make further edits to meet your requirements.

---

> ### Author Response · Authors · 2024-11-30
> **Response to Weakness 5 - Comparing with global prototype distillation**
>
> The short answer is no, the ideas of global anchors in FedLGD and global prototype distillation [2] are not similar. The ideas diverge on (1) how the global anchor/prototype are generated and (2) how they are used to resolve heterogeneity issues.
>
> Notably, the method in [2] does not employ dataset distillation but rather focuses on knowledge distillation, and its global prototype is the aggregated locally shared features, which aligns more closely with frameworks like FedProto [r2], as noted in our Related Work section. Then, [2] uses the global prototype along with knowledge distillation loss to align the **model output logits** to reduce heterogeneity. However, FedLGD relies on dataset distillation to synthesize global anchors from the shared gradient information, and we utilize the global anchor and the supervised contrastive loss to align the clients’ **feature space** to the global anchors’. While knowledge distillation addresses heterogeneity through prototype sharing, it represents an orthogonal strategy to our data-distillation-based approach, making a direct comparison less relevant.
>
> [2] Wu S, Chen J, Nie X, Wang Y, Zhou X, Lu L, Peng W, Nie Y, Menhaj W. Global prototype distillation for heterogeneous federated learning. Scientific Reports. 2024 May 27;14(1):12057.

---

> ### Author Response · Authors · 2024-11-30
> **Response to Weakness 5 - Additional Comments and Questions about IPC**
>
> Images Per Class (IPC) is a commonly used term in the image-based dataset distillation literature, particularly for classification tasks. It refers to the number of virtual images distilled for each class. Since the total size of a distilled dataset is determined by the product of the number of classes and the IPC ($num_{class} \times IPC$), IPC serves as a convenient metric to express the size of the distilled dataset and to define the optimization space for distillation.

---

### Decision · Action_Editor_h4Hh · 2024-12-23

**Recommendation:** Accept as is

**Comment:**

This paper proposed a new method for federated learning on virtual heterogeneous data, using local-global data distillation. All reviewers find the studied setting novel and the results provide new insights. The authors’ rebuttal has successfully addressed the major concerns of reviewers, and all reviewers are in favor of accepting this paper. The reviewers also provided some additional feedback to improve the final version, as listed below
- The main motivation that "distilled virtual data increases heterogeneity" has only weak evidence and needs more justification
- The method reconstructs the virtual data through Federated Gradient Matching, but the clients could simply share the virtual data directly. Neither privacy or dataset size are arguments against this direct approach. The motivation needs to be further discussed.

Overall, I recommend acceptance of this submission. I also expect the authors to include the new results and suggested changes during the rebuttal phase to the final version.

**Audience:**

Of broad interest to federated learning

**Claims And Evidence:**

The claims of are supported by the empirical results.